# Best of Both Worlds: Multimodal Reasoning and Generation via Unified Discrete Flow Matching

**Onkar Susladkar** [1]  **Tushar Prakash** [2]  **Gayatri Deshmukh** [1]  **Kiet A. Nguyen** [1]  **Jiaxun Zhang** [1]  **Adheesh Juvekar** [1]
**Tianshu Bao** [3]  **Lin Chai** [3]  **Sparsh Mittal** [4]  **Inderjit Dhillon** [3,5]  **Ismini Lourentzou** [1]

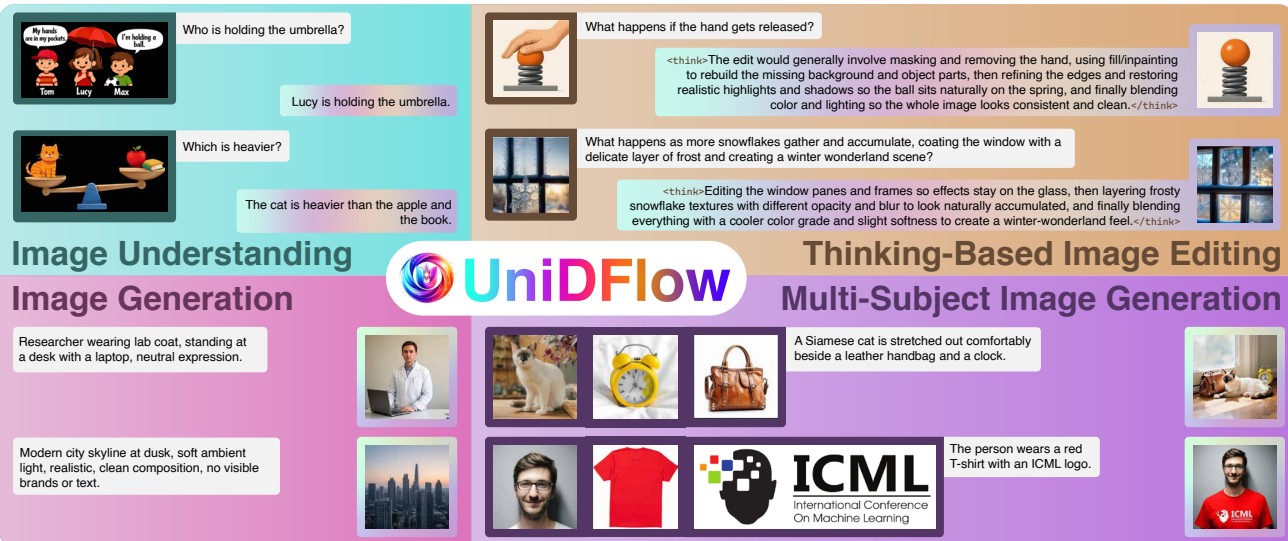

*Figure 1.* We propose UniDFlow, an unified multimodal diffusion framework that supports image understanding, generation, and thinking-based editing. The model performs visual reasoning for question answering, produces high-quality text-to-image generations across diverse scenes and subjects, and enables instruction-driven, multi-step image editing through structured reasoning.

## Abstract

We propose UniDFlow, a unified discrete flow-matching framework for multimodal understanding, image generation, and instruction-guided editing. UniDFlow decouples understanding and generation via task-specific low-rank adapters, avoiding objective interference and representation entanglement, while a novel reference-based multimodal preference alignment optimizes relative outcomes under identical conditioning, improving faithfulness and controllability without large-scale retraining. UniDFlow achieves SOTA performance across six benchmarks and exhibits strong zero-shot generalization to tasks including inpainting, in-context image generation, reference-based editing, and compositional generation, despite no explicit task-specific training.

🖒 PLAN Lab https://plan-lab.github.io/unidflow

## 1. Introduction

Multimodal generative systems have become central to everyday productivity, with large language models (LLMs) such as ChatGPT and Gemini (Team et al., 2023) enabling strong reasoning and instruction following. Similarly, diffusion-based models such as Stable Diffusion (Rombach et al., 2022; Esser et al., 2024) and DALL·E (Ramesh et al., 2021; Betker et al., 2023) excel at high-fidelity image and video generation. However, these models remain largely disjoint as LLM-centric models excel at understanding but lack native generative mechanisms, while diffusion models provide powerful generation with limited semantic grounding and reasoning. This separation motivates unified multimodal models that integrate LLM-level understanding with diffusion-level generation within a single architecture (Wang et al., 2024; Xie et al., 2024).

[1]University of Illinois Urbana-Champaign [2]Sony Research, India [3]Google Research [4]Indian Institute of Technology, Roorkee [5]University of Texas at Austin. Correspondence to: Onkar Susladkar <onkarks2@illinois.edu, lourent2@illinois.edu>.

*Proceedings of the 43rd International Conference on Machine Learning*, Seoul, South Korea. PMLR 306, 2026. Copyright 2026 by the author(s).

Early approaches in this direction, such as Emu (Dai et al., 2023) and Chameleon (Chameleon Team, 2024), model both text and vision using a single auto-regressive (AR) transformer (Vaswani et al., 2017). While simple, AR-based generation is highly inefficient for high-dimensional visual outputs. Hybrid models, including EMMA (He et al., 2025), OmniGenV2 (Wu et al., 2025b), MammothModa2 (Shen et al., 2025), and BAGEL (Deng et al., 2025), combine AR modeling for text with diffusion-style objectives for images to retain language understanding while improving generation. UniDisc (Swerdlow et al., 2025) and MUDDIT (Shi et al., 2025) employ fully discrete diffusion with a unified denoising objective for text and images, but performance lags behind hybrid models.

Despite recent progress, existing unified models still face several fundamental limitations: **(1)** Large-scale AR–diffusion frameworks couple cross-entropy decoding with diffusion-style regression (Shen et al., 2025; Wu et al., 2025b), creating mismatched objectives that lead to unstable joint optimization. **(2)** Even with strong pretrained initialization, many approaches rely on full-model updates over hundreds of millions of samples (Deng et al., 2025; He et al., 2025), incurring substantial compute while often degrading general-purpose reasoning ability. **(3)** Current unified diffusion approaches entangle understanding and generation within shared parameters, thus improving one capability can inadvertently erode the other (Zhong et al., 2026; Shi et al., 2025). **(4)** Generation and editing are often improved through additional alignment stages, such as multimodal reflection (Wu et al., 2025b) or reinforcement learning with scalar rewards (Shen et al., 2025). However, these approaches optimize outputs in isolation, encouraging higher scores or improved reasoning trajectories without modeling relative preference under identical conditioning. As a result, they fail to learn explicit decision boundaries between faithful and subtly incorrect edits.

To address the aforementioned limitations, we introduce UniDFlow, a unified discrete diffusion framework for efficient multimodal understanding and generation. UniDFlow leverages a strong pretrained vision–language model as a prior, avoiding redundant pretraining and enabling parameter-efficient adaptation through lightweight adapters. We perform large-scale three-stage training: (i) an understanding-focused stage, (ii) a generation-focused stage, and (iii) a joint understanding–generation stage with reference-based multimodal preference optimization to improve editing fidelity and controllability. To prevent parameter entanglement, UniDFlow trains separate adapters for understanding and generation, while the final stage trains only a lightweight router to combine them dynamically.

Fig. 2 visualizes the instruction-guided activation maps during editing. UniDFlow consistently attends more precisely

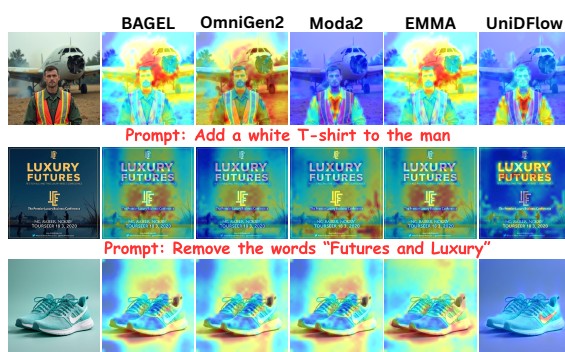

Figure 2. Instruction-guided editing attention maps showing UniD-Flow more precisely focuses on relevant regions than prior models.

to instruction-relevant regions, whether modifying coarse objects (*e.g.*, adding a T-shirt) or finer details (*e.g.*, changing the swoosh color). Our main contributions are:

- We introduce UniDFlow, a unified discrete diffusion model that repurposes a pretrained vision–language backbone as a generator over multimodal tokens, enabling understanding, text generation, image synthesis, and editing within one probabilistic interface.

- We unify text and image generation under a *single discrete flow-matching objective* for all tasks and incorporate a stable time-conditioning mechanism that preserves the backbone's reasoning priors. Compared to prior multi-objectives, UniDFlow achieves efficient training and inference, requiring only 20 denoising steps while preserving high generation quality.

- We propose mRef-DPO, a *reference-guided multimodal preference alignment* that directly optimizes a pairwise log-likelihood margin between preferred and non-preferred outcomes, anchored to a frozen reference model, leading to more faithful and controllable editing behavior and stabilized training by enforcing comparative preference learning and rewarding improvements over the base model rather than absolute score maximization.

- UniDFlow achieves state-of-the-art performance on eight benchmarks spanning understanding, generation, and editing, with up to 13% improvement over larger unified models with more than 3× parameters, and up to 24% gains over popular models such as Qwen 3 (Bai et al., 2025b) and DeepSeek-VL2 (Wu et al., 2024b).

## 2. Related Work

**Diffusion for Visual Generation.** Diffusion probabilistic models (DPMs) (Ho et al., 2020; Nichol et al., 2021; Saharia et al., 2022) outperform GANs (Goodfellow et al., 2014) in stability and quality but are costly in pixel space. Latent diffusion models (LDMs) (Rombach et al., 2022) mitigate this via compressed latent representations, enabling strong text-to-image generation (Zhang et al., 2023; Chen et al., 2024a; Podell et al., 2023). Discrete diffusion (Austin

et al., 2021) extends diffusion to categorical spaces using masking-based corruption, motivating parallel mask-and-predict generators that improve fidelity and efficiency (Gu et al., 2022; Chang et al., 2022).

**Unified Models for Understanding and Generation.** To unify understanding and generation, early works such as Emu (Dai et al., 2023; Sun et al., 2024) and Chameleon (Chameleon Team, 2024) adopt fully autoregressive modeling over text and visual tokens, but scale poorly for high-resolution images. Hybrid frameworks including EMMA (He et al., 2025), OmniGenV2 (Wu et al., 2025b), MammothModa2 (Shen et al., 2025), and BAGEL (Deng et al., 2025) combine autoregressive text modeling with diffusion-based image generation, yet still face modality and objective mismatches. Fully discrete diffusion models such as UniDisc (Swerdlow et al., 2025) and MUDDIT (Shi et al., 2025) further unify modeling but lag behind large-scale hybrids. Our work introduces UniDFlow, a unified discrete flow-matching model with stable time-conditioning that preserves reasoning priors and enables efficient, high-fidelity multimodal generation and editing.

**LLM and Diffusion Preference Alignment.** LLMs (Touvron et al., 2023; Liu et al., 2024a) provide strong reasoning with autoregressive Transformers, and VLMs (Bai et al., 2023; 2025a) extend them to images by projecting visual features (*e.g.*, SigLIP (Zhai et al., 2023)) into the language token space. Qwen (Bai et al., 2025b), LLaVA (Liu et al., 2023), BLIP-2 (Li et al., 2023), and Flamingo (Alayrac et al., 2022) excel at multimodal understanding but typically rely on separate diffusion backbones for image generation and editing. Preference learning has also been adapted to diffusion models, including Diffusion-DPO (Wallace et al., 2024), score-space alignment (DSPO) (Zhu et al., 2025), and stabilized variants such as DGPO (Luo et al., 2025) and discrete-diffusion extensions (Borso et al., 2025). Prior work further improves controllability via additional alignment stages (*e.g.*, multimodal reflection (Wu et al., 2025b) or scalar-reward RL (Shen et al., 2025)). In contrast, UniD-Flow performs *reference-based multimodal preference alignment*, optimizing a pairwise log-likelihood margin against a frozen reference model for stable, comparative supervision, improving faithfulness and controllable editing.

# 3. Method

## 3.1. Preliminaries: Discrete Flow Matching

We use Discrete Flow Matching (DFM) (Gat et al., 2024) as the common objective across all training stages. DFM learns a transport field in discrete spaces by mapping samples from noise to data. Let $x_0 \sim q_{\text{data}}$ denote a clean discrete sample (*e.g.*, text or visual tokens), and $x_t$ its corrupted version at time step $t \in \{0, \ldots, T\}$ generated by a fixed forward

noising process $q(x_t \mid x_0, t)$. Given $x_t$, a flow network $f_\theta(x_t, t, c)$ conditioned on time $t$ and context $c$ predicts the transport toward the clean state as $f_\theta(x_t, t, c) \approx q(x_0 \mid x_t, t, c)$. The model is trained by minimizing a token-wise categorical negative log-likelihood:

$$\mathcal{L}_{\text{DFM}}(\theta; x_0 \mid x_t, t, c) = \mathbb{E}_{x_0, t, x_t} \left[ -\log f_\theta(x_0 \mid x_t, t, c) \right]. \quad (1)$$

At inference, sampling starts from $x_T \sim q_{\text{noise}}$ and applies the learned flow to recover $x_0$. By directly estimating transport directions, DFM enables efficient few-step sampling, with conditioning via $c$ supporting unified language modeling, visual generation, and editing.

## 3.2. UniDFlow

We cast multimodal understanding, conditional generation, and instruction-based image editing as a single discrete denoising process. Starting from a pretrained vision–language transformer with parameters $\theta_0$, UniDFlow learns to recover a clean token sequence from a corrupted one under appropriate conditioning. For understanding, the denoised sequence corresponds to answer text tokens conditioned on an instruction $p$ and an input image $x$; for generation and editing, it corresponds to visual tokens conditioned on $p$ and a reference image $x_{\text{ref}}$. To enable discrete diffusion over images, we map images to sequences of discrete visual tokens using a pretrained tokenizer, and we use bidirectional self-attention to support full-context denoising. All task-specific adaptation is implemented with low-rank adapters (LoRA), while $\theta_0$ remains frozen.

Our training follows a three-stage pipeline (illustrated in Figs. 3 and 4): Stage I aligns the pretrained vision–language backbone for diffusion-based multimodal understanding, Stage II adapts the model for discrete visual generation while preserving reasoning capabilities, and Stage III performs reference-based multimodal preference alignment to improve fidelity and controllability. We first describe the time-conditioned normalization used throughout the model, followed by the three training stages.

### 3.2.1. TIME-STEP GUIDED RMSNORM

Conditioning a pretrained transformer on diffusion time by directly adding time embeddings to attention or MLP activations can destabilize training by perturbing learned feature distributions. We address this with Time-Step Guided RMSNorm (TSG-RMSNorm), which injects time information by modulating the RMSNorm scale parameters rather than altering the activations themselves. This preserves pretrained representations by keeping the direction of hidden states unchanged while only applying a controlled, time-dependent rescaling.

Let $h_\ell \in \mathbb{R}^d$ denote the input hidden state (activation vector) to the RMSNorm layer at transformer layer $\ell$. Stan-

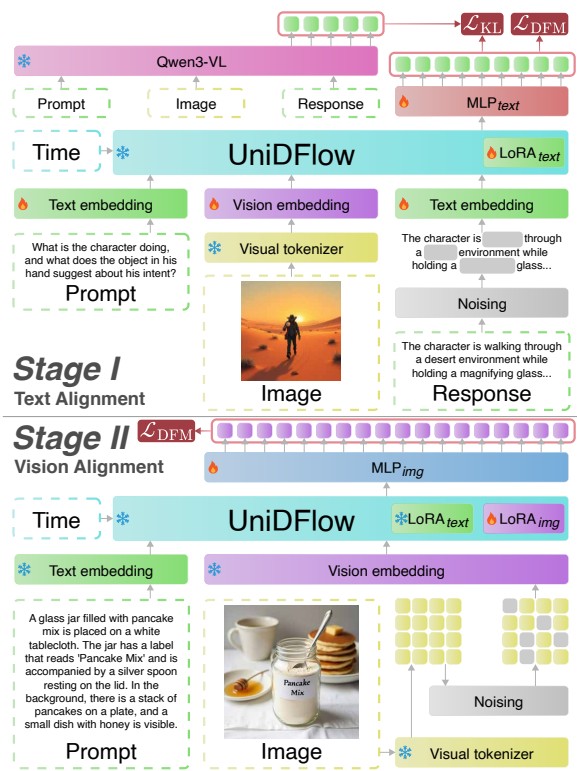

*Figure 3.* Overview of Stage I (understanding via text alignment) and Stage II (generation via vision alignment) of UniDFlow.

dard RMSNorm is $\text{RMSNorm}(h_\ell) = \gamma_\ell \odot \frac{h_\ell}{\text{RMS}(h_\ell)}$, where $\text{RMS}(h_\ell) = \sqrt{\frac{1}{d} \sum_{j=1}^{d} h_{\ell,j}^2 + \varepsilon}$. Given a time embedding $e(t)$, we predict a time-dependent modulation for each layer, *i.e.*, $s_\ell(t) = W_\ell^{(s)} e(t)$, $b_\ell(t) = W_\ell^{(b)} e(t)$. We apply these to the pretrained RMSNorm parameters via

$$\begin{aligned} \text{TSG-RMSNorm}(h_\ell, t) &= \text{RMSNorm}(h_\ell) \\ &\odot \big( \gamma_\ell \odot (1 + s_\ell(t)) \big) + b_\ell(t), \end{aligned} \quad (2)$$

where $\gamma_\ell$ is the pretrained RMSNorm scale and $\odot$ denotes element-wise multiplication. All time-modulation parameters are zero-initialized so that $s_\ell(t) = 0$ and $b_\ell(t) = 0$ at initialization, exactly recovering the pretrained model.

### 3.2.2. STAGE I: TEXT ALIGNMENT

Unified multimodal models often entangle understanding and generation objectives, leading to representational interference and degraded reasoning. We first adapt the pretrained backbone to diffusion-style understanding through text alignment in isolation, preserving language–visual reasoning before introducing generative training.

Given an instruction $p$, visual tokens $x$, and a fully masked text token sequence $y_{\text{txt},t}$, the model predicts the clean answer tokens $y_{\text{txt},0}$ using discrete flow matching. The training objective follows Eq.( 1):

$$\mathcal{L}_{\text{under}} = \mathcal{L}_{\text{DFM}}\big( \Delta\theta_u \,;\, y_{\text{txt},0} \mid y_{\text{txt},t},\, p,\, x \big). \quad (3)$$

where $\theta_0$ denotes frozen pretrained VLM parameters and $\Delta\theta_u$ are $LoRA_{text}$ adapters specialized for understanding.

To prevent semantic drift from the pretrained language behavior, we additionally regularize the diffusion-predicted distribution with a KL divergence against the autoregressive answer distribution produced by the original VLM:

$$\mathcal{L}_{\text{KL}} = \text{KL}\big( p_{\text{DFM}}(y_{\text{txt},0} \mid y_{\text{txt},t},\, t,\, p,\, x) \,\|\, p_{\text{AR}}(y_{\text{txt}} \mid p,\, x) \big). \quad (4)$$

This constraint anchors diffusion-based decoding to the pretrained linguistic manifold while allowing bidirectional attention and time-conditioned normalization to support non-autoregressive reasoning.

The total Stage I objective is $\mathcal{L}_{\text{Stage I}} = \mathcal{L}_{\text{under}} + \lambda_{\text{KL}} \mathcal{L}_{\text{KL}}$.

### 3.2.3. STAGE II: VISION ALIGNMENT

This stage adapts the same frozen backbone for conditional generation in discrete visual token space, while preserving the understanding behavior learned in the previous training stage. We keep $\theta_0$ and $\Delta\theta_u$ frozen and introduce a separate set of LoRA adapters $\Delta\theta_g$ specialized for generation.

Given an instruction $p$ and corrupted visual tokens $y_{\text{vis},t}$, the model predicts clean visual tokens $y_{\text{vis},0}$ using discrete flow matching with parameters $\theta_0 + \Delta\theta_u + \Delta\theta_g$:

$$\mathcal{L}_{\text{Stage II}} = \mathcal{L}_{\text{DFM}}\big( \Delta\theta_g \,;\, y_{\text{vis},0} \mid y_{\text{vis},t},\, t,\, p \big), \quad (5)$$

where only $\Delta\theta_g$ ( $LoRA_{img}$) is trainable, while $\theta_0$ and the understanding adapters $\Delta\theta_u$ are kept frozen. The diffusion process operates entirely in a discrete latent space, enabling efficient sampling and seamless integration with the backbone's token-based architecture. By isolating generation-specific parameters, Stage II establishes strong conditional image generation capabilities without interfering with the language and reasoning behavior learned during Stage I.

### 3.2.4. STAGE III: REFERENCE-BASED MULTIMODAL PREFERENCE ALIGNMENT

While the previous stages endow UniDFlow with strong multimodal understanding and generation capabilities, token-level likelihood training cannot reliably distinguish between multiple plausible outputs that differ in instruction fidelity, visual grounding, or reasoning consistency. Stage III therefore introduces a reference-based multimodal preference alignment objective that explicitly optimizes relative preferences across text, vision, and reflection, grounded in reference images.

Each preference instance specifies an instruction $p$ with paired preferred($w$)/rejected($l$) outcomes: reference image $(x_{\text{ref}}^w, x_{\text{ref}}^l)$, text responses $(y_{\text{txt}}^w, y_{\text{txt}}^l)$, visual tokens $(y_{\text{vis}}^w, y_{\text{vis}}^l)$, and reflection sequences $(r^w, r^l)$, allowing the model to learn which multimodal outcomes are preferred, conditioned on both the instruction and the visual reference.

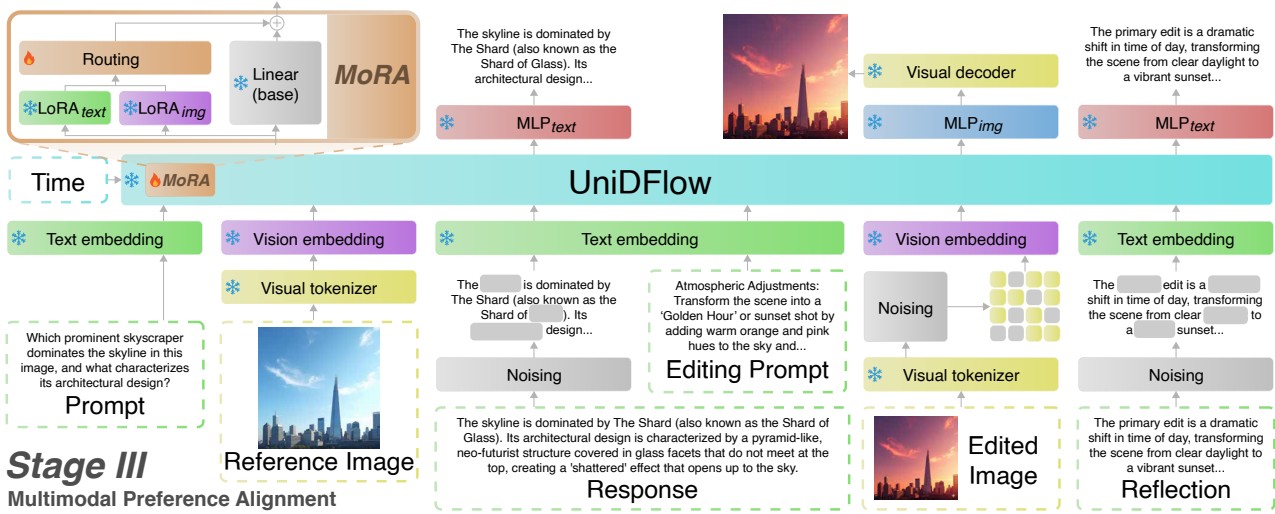

*Figure 4.* Stage III of UniDFlow: reference-based multimodal preference alignment for improved faithfulness, controllability, and editing.

**Mixture-of-LoRA Routing (MoRA).** Since this stage optimizes preferences for both understanding and generation, naively sharing parameters can introduce objective interference, while static routing restricts adaptability. Therefore, we learn a lightweight router $r_\phi$ with parameters $\phi$ that dynamically composes task-specific adapters based on the hidden state at diffusion step $t$:

$$\Delta\theta(t) = \alpha_t \Delta\theta_u + (1-\alpha_t)\Delta\theta_g, \quad \alpha_t = r_\phi(h_t), \quad (6)$$

**Multimodal Preference Learning.** We adopt a reference-anchored Direct Preference Optimization (DPO) objective with a frozen reference policy $\pi_{\text{ref}}$. For text, the loss is $\mathcal{L}_{\text{tRef-DPO}} = -\log\sigma\left(\beta\Delta_\theta^{txt}\right)$ and preference margin is

$$\Delta_\theta^{\text{txt}} = \log\frac{\pi_\theta(y_{\text{txt}}^w|p,x_{\text{ref}}^w)}{\pi_{\text{ref}}(y_{\text{txt}}^w|p,x_{\text{ref}}^w)} - \log\frac{\pi_\theta(y_{\text{txt}}^l|p,x_{\text{ref}}^l)}{\pi_{\text{ref}}(y_{\text{txt}}^l|p,x_{\text{ref}}^l)}, \quad (7)$$

For vision, we concatenate reflection and image tokens as $\tilde{y}_{\text{vis}} = (r, y_{\text{vis}})$, similarly $\mathcal{L}_{\text{vRef-DPO}} = -\log\sigma\left(\beta\Delta_\theta^{vis}\right)$.

$$\Delta_\theta^{\text{vis}} = \log\frac{\pi_\theta(\tilde{y}_{\text{vis}}^w|p,x_{\text{ref}}^w)}{\pi_{\text{ref}}(\tilde{y}_{\text{vis}}^w|p,x_{\text{ref}}^w)} - \log\frac{\pi_\theta(\tilde{y}_{\text{vis}}^l|p,x_{\text{ref}}^l)}{\pi_{\text{ref}}(\tilde{y}_{\text{vis}}^l|p,x_{\text{ref}}^l)}. \quad (8)$$

Stage III jointly aligns text and vision through a preference-augmented objective: $\mathcal{L}_{\text{mRef-DPO}} = \lambda_t\mathcal{L}_{\text{tRef-DPO}} + \lambda_v\mathcal{L}_{\text{vRef-DPO}}$, which promotes faithful instruction following, grounded visual editing, and consistent multimodal behavior under reference conditioning.

We optimize a unified objective that combines discrete flow-matching (DFM) likelihood training for three output streams, text generation, $\mathcal{L}_{\text{text}} = \mathcal{L}_{\text{DFM}}\left(\phi; y_{\text{txt},0}^w \mid y_{\text{txt},t}^w, x_{\text{ref}}^w, p, t\right)$, visual editing $\mathcal{L}_{\text{edit}} = \mathcal{L}_{\text{DFM}}\left(\phi; y_{\text{vis},0}^w \mid y_{\text{vis},t}^w, x_{\text{ref}}^w, p, t\right)$, and reflection, with reference-anchored multimodal preference alignment $\mathcal{L}_{\text{refl}} = \mathcal{L}_{\text{DFM}}\left(\phi; r_0^w \mid r_t^w, x_{\text{ref}}^w, y_{\text{vis},t}^w, y_{\text{txt},t}^w, p, p_{\text{edit}}, t\right)$. Thus, the final objective for Stage III is:

$$\mathcal{L}_{\text{Stage-III}} = \mathcal{L}_{\text{text}} + \mathcal{L}_{\text{edit}} + \mathcal{L}_{\text{refl}} + \mathcal{L}_{\text{mRef-DPO}} \quad (9)$$

The DFM terms maximize time-conditioned token likelihood along the discrete diffusion trajectory under their respective conditionings (instruction, reference image, or edit prompt), enforcing token-level consistency. The $\mathcal{L}_{\text{mRef-DPO}}$ term introduces comparative alignment by increasing the log-likelihood margin of preferred over rejected outputs relative to a frozen reference policy $\pi_{\text{ref}}$, stabilizing training and improving cross-modal faithfulness.

## 4. Experiments

We conduct extensive experiments to evaluate the performance of UniDFlow across six benchmarks, covering multimodal understanding, generation, and editing. In Stage I, we train using MMINSTRUCT (Liu et al., 2024b) to establish strong multimodal understanding. Stage II focuses on generative capability by training on TEXT-TO-IMAGE-4M (Jacky-hate, 2024; Sun et al., 2023; Schuhmann et al., 2022) . Stage III performs reference-based multimodal preference alignment with 3.5M curated preference samples under identical inputs and reference images. Dataset construction for preference alignment, training, and implementation details are provided in Appendices A-B.

### 4.1. Multi-Modal Understanding

Table 1 reports results on the EVALVLM benchmark. Compared to strong unified hybrid baselines such as BAGEL (7B MoT), UniDFlow achieves a +6.9% improvement on MME-P and +7.0% on MME-S, indicating stronger perceptual and reasoning consistency. Against EMMA (4B), UniDFlow further improves MMBench by +6.3% and Math-Vista by +13.3%, demonstrating superior mathematical and multi-step reasoning despite comparable model scale. Moreover, compared to the unified diffusion baseline MUDDIT, UniDFlow achieves an overall improvement of 12% across

*Table 1.* Comparison of multimodal understanding performance on EVALVLMBENCH (Fu et al., 2023; Liu et al., 2024c; Yue et al., 2024; Yu et al., 2024c; Lu et al., 2024; Tong et al., 2024) across diverse reasoning tasks.

| Model | Params | MME-P | MME-S | MMBENCH | MMMU | MM-VET | MATHVISTA | MMVP |
|---|---|---|---|---|---|---|---|---|
| Qwen2.5-VL (Bai et al., 2025a) | 3B | – | 2157 | 79.1 | 53.1 | 61.8 | 62.3 | – |
| BLIP-3 (Xue et al., 2024) | 4B | – | – | 76.8 | 41.1 | – | 39.6 | – |
| DeepSeek-VL2 (Wu et al., 2024b) | 4B | – | – | 51.1 | 60.0 | 62.8 | – | – |
| Qwen3-VL (Bai et al., 2025b) | 4B | – | – | 85.1 | 64.1 | 72.5 | – | – |
| VILA-U (Wu et al., 2024a) | 7B | 1336 | – | 66.6 | 32.2 | 27.7 | – | 22.0 |
| Chameleon (Chameleon Team, 2024) | 7B | – | – | 35.7 | 28.4 | 8.3 | – | 0.0 |
| Janus-Pro (Chen et al., 2025a) | 7B | 1567 | – | 79.2 | 41.0 | 50.0 | – | – |
| TokenFlow-XL (Geyer et al., 2023) | 13B | 1546 | – | 68.9 | 38.7 | 40.7 | – | – |
| BAGEL (Deng et al., 2025) | 7B | 1687 | 2388 | 85.0 | 55.3 | 67.2 | 73.1 | 69.3 |
| OmniGenV2 (Wu et al., 2025b) | 8B | – | – | 53.1 | 61.5 | – | – | – |
| EMMA (He et al., 2025) | 4B | – | – | 85.8 | 65.1 | 73.0 | 75.8 | – |
| MammothModa-2 (Shen et al., 2025) | 4B | 1753 | 1998 | 86.6 | 71.23 | 79.4 | 81.8 | 77.5 |
| MIDDIT (Shi et al., 2025) | 4B | 1700 | 1832 | 82.8 | 66.6 | 76.2 | 79.1 | 74.1 |
| **UniDFlow** | 4B | **1803** | **2555** | **91.2** | **74.3** | **82.7** | **85.9** | **80.2** |

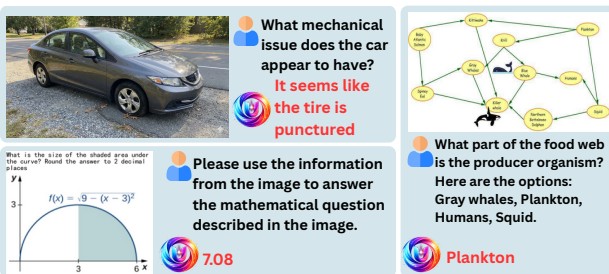

*Figure 5.* Multimodal reasoning from UniDFlow

different understanding tasks. Finally, when compared with leading understanding-only models such as Qwen2.5-VL (3B), UniDFlow attains 20.4% higher overall performance, underscoring the effectiveness of its pretrained VLM prior, bidirectional attention, and discrete diffusion formulation for multimodal understanding. Additional results on OCR-BENCHV2 (Fu et al., 2024) can be found in Appendix E. Fig. 5 shows reasoning-based text generation examples, where UniDFlow can accurately extract information from images to respond to user queries.

## 4.2. Text-to-Image Generation

Table 2 summarizes the performance of UniDFlow on GENEVAL and DPGBENCH for multimodal generation. On GENEVAL, which evaluates compositional text-to-image generation across object counting, attribute binding, and spatial reasoning, UniDFlow achieves an overall score of 0.95, outperforming strong unified baselines such as EMMA and MammothModa2 by +2.2% and +9.2%, respectively, highlighting its stronger ability to associate attributes with the correct objects under compositional constraints. A similar trend is observed on DPGBENCH, which evaluates fine-grained prompt grounding across global understanding, attribute binding, and relational reasoning, where UniDFlow outperforms EMMA and MammothModa2 by +6.5% and +4.6%, respectively. Notably, UniDFlow also surpasses generation-focused models such as Qwen-Image (7B+20B)

*Table 2.* Overall generation performance on GENEVAL (Ghosh et al., 2023) and DPGBENCH (Hu et al., 2024). Full benchmark-wise breakdowns are provided in Appendix C.

| Model | Params | GenEval | DPGBench |
|---|---|---|---|
| DALL-E 3 (Betker et al., 2023) | – | 0.67 | 83.50 |
| SD3-Medium (Esser et al., 2024) | 2B | 0.74 | 80.43 |
| Qwen-Image(-RL) (Wu et al., 2025a) | 7B+20B | 0.91 | 88.32 |
| TokenFlow-XL (Geyer et al., 2023) | 14B | 0.55 | – |
| Janus-Pro-7B (Chen et al., 2025a) | 7B | 0.80 | 84.19 |
| BAGEL (Deng et al., 2025) | 7B+7B | 0.88 | 87.74 |
| OmniGenV2 (Wu et al., 2025b) | 3B+4B | 0.78 | 83.57 |
| MammothModa-2 (Shen et al., 2025) | 8B+3B+2B | 0.87 | 87.20 |
| EMMA (He et al., 2025) | 4B | 0.93 | 85.63 |
| MUDDIT (Shi et al., 2025) | 8B | 0.90 | 86.37 |
| **UniDFlow** | 4B | **0.95** | **91.19** |

by 4.0% on GENEVAL and 3.2% on DPGBENCH, despite using substantially fewer parameters. Fig. 6 further demonstrates that UniDFlow produces visually faithful and prompt-consistent images, accurately rendering fine-grained details and background structures, which reflect strong global semantics and local visual fidelity.

## 4.3. Text-to-Image Editing

Table 3 summarizes the image editing performance of UniDFlow on IMGEDIT BENCH (Ye et al., 2025), EMU-EDIT (Sheynin et al., 2024), and GEDIT-BENCH-EN (Liu et al., 2025). On EMU-EDIT, UniDFlow outperforms EMMA and MammothModa2 by approximately +3.5% and +4.1%, respectively, indicating stronger semantic alignment between the input image, editing instruction, and edited output. On GEDIT-BENCH-EN, which emphasizes perceptual quality and instruction satisfaction, UniDFlow improves the averaged score by +3.7% over EMMA and +2.9% over MammothModa2. Further, on IMAGEEDIT BENCH, which evaluates diverse editing scenarios including object manipulation, background changes, style transfer, and hybrid edits, UniDFlow achieves an overall score of 4.24, surpassing EMMA (4.01) and MammothModa2 (4.06) by +5.7% and +4.4%, respectively. Notably, the largest gains are observed in Extract and Remove operations, demonstrating

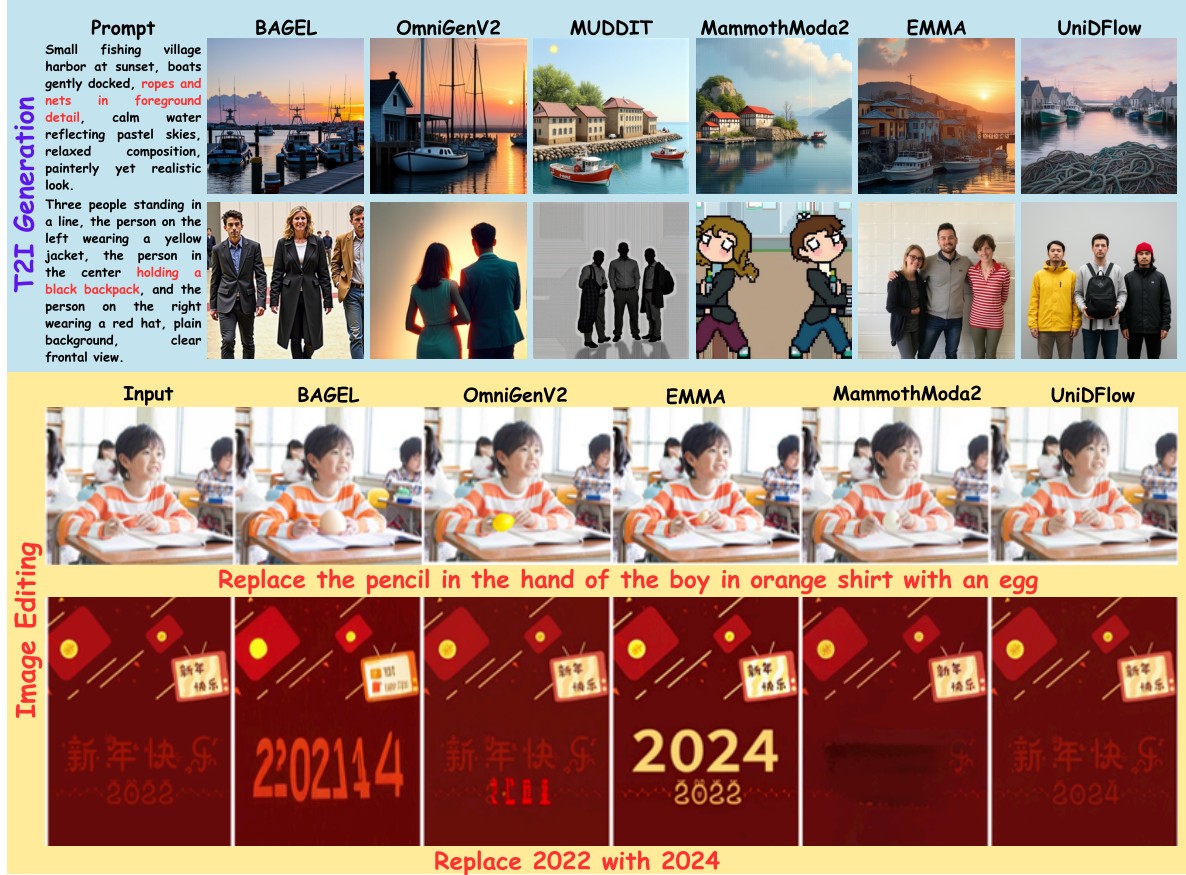

*Figure 6.* Qualitative comparison of compositional text-to-image generation and editing. Prompts require precise grounding of attributes and spatial relations (red text). UniDFlow consistently adheres to these constraints while maintaining realistic structure and visual fidelity, outperforming prior unified baselines in fine-grained prompt alignment. More results can be found in Appendix D.

*Table 3.* Text-to-image editing results. IMGEDIT metric are category-wise scores, while EMU-EDIT CLIP-I/DINO are used for source consistency and CLIP-Out for caption alignment. GEDIT-BENCH-EN evaluates SC (instruction following) and PQ (perceptual quality).

| Model | IMGEDIT | | | | EMU-EDIT | | | GEDIT-BENCH-EN | | |
|---|---|---|---|---|---|---|---|---|---|---|
| | Add ↑ | Extract ↑ | Remove↑ | Overall ↑ | CLIP-I ↑ | CLIP-Out ↑ | DINO ↑ | SC ↑ | PQ ↑ | Overall ↑ |
| FLUX.1 Kontext-Pro (Labs et al., 2025) | 4.25 | 2.35 | 3.57 | 4.00 | 0.88 | - | 0.808 | 7.77 | 7.12 | 6.95 |
| BAGEL (Deng et al., 2025) | 3.56 | 1.70 | 2.62 | 3.20 | 0.839 | 0.307 | 0.753 | 7.36 | 6.83 | 6.52 |
| UniWorld-v1 (Lin et al., 2025) | 3.82 | 2.27 | 3.24 | 3.26 | – | – | – | 4.93 | 7.43 | 4.85 |
| OmniGenV2 (Wu et al., 2025b) | 3.57 | 1.77 | 3.20 | 3.44 | 0.876 | 0.309 | 0.822 | 7.16 | 6.77 | 6.41 |
| Emma (He et al., 2025) | 4.52 | 3.54 | 4.21 | 4.01 | 0.911 | 0.311 | 0.834 | 7.33 | 7.54 | 6.52 |
| MammothModa2 (Shen et al., 2025) | 4.57 | 3.38 | 3.34 | 4.06 | 0.891 | 0.322 | 0.844 | 7.77 | 7.32 | 6.82 |
| **UniDFlow** | **4.66** | **4.01** | **4.24** | **4.24** | **0.921** | **0.362** | **0.862** | **8.01** | **7.82** | **7.12** |

more precise target isolation and reduced collateral degradation. These improvements are driven by reference-based preference alignment, which encourages UniDFlow to select higher-quality edits that better satisfy user intent.

**Editing with reasoning.** Fig. 6 presents image editing qualitative examples, where UniDFlow produces both accurate, large-scale semantic edits (*e.g.*, style transfer) and fine-grained object-level modifications, exhibiting strong instruction fidelity and precise edit localization.

Fig. 7 further compares models on reasoning-driven editing tasks requiring temporal, geometric, and physical reasoning. UniDFlow generates outputs that better reflect the intended transformations while preserving object identity, benefiting from the strong reasoning priors inherited from the pretrained VLM backbone.

**Subject-driven generation.** Furthermore, UniDFlow supports in-context subject-driven image generation from multiple reference images, as shown in Fig. 8, without any explicit task-specific training. Given reference images and a textual instruction, UniDFlow synthesizes a coherent output while preserving fine-grained visual details from the references. This behavior emerges from its unified multimodal optimization, which enables joint reasoning over object identity, attributes, and spatial relations.

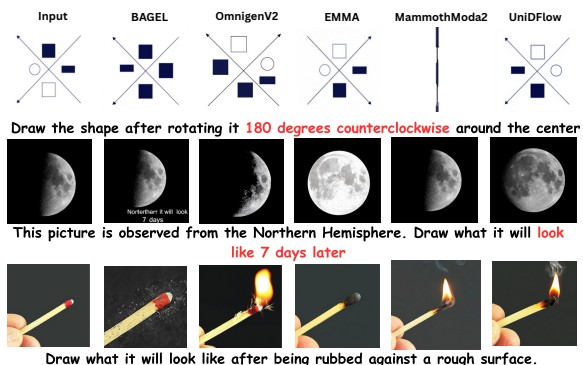

*Figure 7.* Reasoning-driven image editing, highlighting temporal, geometric, and physical transformations handled by UniDFlow.

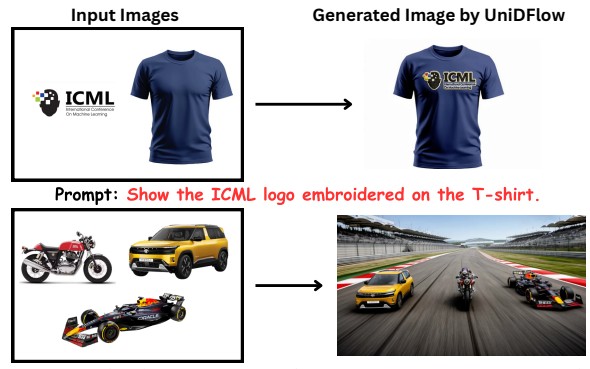

*Figure 8.* Subject-driven image generation with attribute editing and multi-object compositional reasoning.

**Analysis on editing complexity.** To understand how reflection affects edits of varying difficulty, we conduct a complexity-controlled analysis on 200 images from PIE-Bench (Ju et al., 2024), covering add/remove, reasoning-based, scene-text, and multi-object edits. Since reflection is generated in parallel with the edited output, this analysis isolates whether it provides useful localization and preservation cues across different edit types. As shown in Table 4, reflection consistently improves PSNR and SSIM, reduces structural distance, and increases CLIP similarity for both edited and preserved regions. The strongest gains appear on reasoning-based and multi-object edits, indicating that reflection is especially useful when the model must localize the correct region under more complex instructions.

## 4.4. Ablations

Table 5 presents a comprehensive ablation study analyzing the key design choices of UniDFlow.

**Model sizes.** Performance improves consistently as model size increases across all benchmarks. Larger backbones provide stronger multimodal priors and improved capacity for modeling long-range dependencies, which benefits both reasoning and diffusion-based generation. Notably, even the 4B model achieves competitive performance, validating the parameter-efficient design of UniDFlow.

*Table 4.* Analysis of the reflection mechanism across editing tasks with varying complexity.

| Setting | Type | PSNR↑ | SSIM↑ | SD↓ | CLIP$_{ed}$↑ | CLIP$_{un}$↑ |
|---|---|---|---|---|---|---|
| Refl. | Add/Rem. | 31.43 | 0.856 | 9.12 | 26.67 | 25.05 |
| | Reason. | 27.82 | 0.869 | **7.45** | 27.53 | 24.11 |
| | Sc. Text | 30.11 | **0.891** | 10.43 | **31.44** | **27.62** |
| | Multi-Obj. | **31.45** | 0.847 | 9.52 | 28.82 | 25.01 |
| w/o Refl. | Add/Rem. | 29.44 | 0.833 | 11.33 | 24.44 | 23.37 |
| | Reason. | 23.67 | 0.849 | 10.02 | 23.12 | 23.78 |
| | Sc. Text | 27.52 | 0.871 | 12.02 | 27.72 | 25.28 |
| | Multi-Obj. | 28.89 | 0.825 | 11.78 | 25.17 | 21.75 |

**Visual tokenizer.** UniDFlow uses PyraTok (Susladkar et al., 2026), which performs text-guided multi-scale quantization, enabling coarse-to-fine visual representations aligned with language. In contrast, 3D-MBQ-VAE (Susladkar et al., 2024) and MAGVIT-v2 (Yu et al., 2024a) use single-scale, visually trained tokenizers, limiting hierarchical modeling and text alignment. SweetTok (Tan et al., 2025) incorporates text semantics but lacks multi-scale quantization, reducing its ability to capture coarse-to-fine structure.

**Components.** Removing either understanding-specific or generation-specific LoRA adapters leads to noticeable degradation, confirming that separating task-specific adaptations is critical to avoid objective interference. Performance drops further when the router is removed, indicating that dynamic composition of adapters is necessary for balancing understanding and generation. Using a single shared LoRA fails entirely, demonstrating that naive parameter sharing causes severe entanglement between tasks.

**Losses.** Removing visual or text alignment losses degrades performance on corresponding benchmarks. Excluding reflection-based preference learning reduces editing and faithfulness metrics, showing that reasoning behind generation helps in precise instruction following and multimodal editing (refer to Appendix D for visual results).

**Alignment Training.** To assess the effect of our proposed mRef-DPO alingment method, we replace it with DPO (Rafailov et al., 2023) or uni-GRPO (Yang et al., 2025). Vanilla DPO can hurt when text–image tokens are weakly aligned, yielding noisy preference signals that degrade reasoning-grounded generation and edits. Uni-GRPO gives small gains but its group normalization is unstable (especially on short prompts), reducing fine-grained edit reliability. mRef-DPO performs best by using modality-aware preference learning to stabilize cross-modal credit assignment between textual reasoning and diffusion steps, improving alignment and edit precision across metrics.

**Time-step Conditioning.** In-context token concatenation is simple but lacks per-layer granularity, and its conditioning signal is diluted as sequence length grows (up to 4096 tokens in Stage III), degrading DPGBench (−1.10) and ImgEdit

*Table 5.* Ablations. All results are averaged across benchmarks.

| | EvalVLM | GenEval | DPGBench | ImgEdit |
|---|---|---|---|---|
| **UniDFlow** | **82.85** | **0.95** | **91.19** | **4.24** |
| **1. Model Size Ablation** | | | | |
| Qwen3-0.6B | 79.48 | 0.93 | 88.32 | 4.19 |
| Qwen3-4B | 82.85 | 0.95 | 91.91 | 4.24 |
| Qwen3-8B | 84.02 | 0.96 | 92.56 | 4.26 |
| Qwen3-14B | 89.24 | 0.98 | 95.44 | 4.63 |
| **2. Visual tokenizer** | | | | |
| 3D-MBQ-VAE | 81.27 | 0.92 | 91.43 | 4.19 |
| MAGVIT-v2 | 81.19 | 0.91 | 90.34 | 4.16 |
| SweetTok | 80.76 | 0.92 | 90.44 | 4.12 |
| **3. Architectural Ablations** | | | | |
| w/o LoRA$_{text}$ | 80.11 | 0.92 | 89.33 | 4.01 |
| w/o LoRA$_{img}$ | 81.23 | 0.93 | 90.05 | 4.08 |
| w/o MoRA | 80.67 | 0.93 | 89.88 | 4.11 |
| Single LoRA (Und+Gen) | 79.92 | 0.90 | 89.12 | 4.08 |
| **4. Loss Function Ablations** | | | | |
| w/o $\mathcal{L}_{vRef-DPO}$ | 80.45 | 0.91 | 88.44 | 4.09 |
| w/o $\mathcal{L}_{tRef-DPO}$ | 79.45 | 0.91 | 90.32 | 4.18 |
| w/o $\mathcal{L}_{mRef-DPO}$ | 77.34 | 0.86 | 86.23 | 4.05 |
| w/o Reflection | 81.23 | 0.89 | 87.57 | 4.14 |
| **5. Stage-III Alignment Training** | | | | |
| DPO | 80.12 | 0.92 | 88.82 | 4.14 |
| uni-GRPO | 80.88 | 0.93 | 90.07 | 4.18 |
| **6. Time Step Conditioning** | | | | |
| In-Context with tokens | 82.11 | 0.94 | 90.09 | 4.15 |
| AdLN | 81.53 | 0.93 | 88.75 | 4.0 |

$(-0.09)$. AdLN underperforms as it overrides the frozen Qwen3-VL backbone's learned normalization statistics, disrupting its reasoning priors. In contrast, TSGN-RMSNorm modulates the existing RMSNorm affine parameters via zero-initialized multiplicative $(1 + s_\ell(t))$ and additive $b_\ell(t)$ terms, recovering the pretrained mapping exactly at $t=0$. This preserves the pretrained hidden-state geometry while injecting per-layer, timestep-dependent conditioning, with negligible overhead beyond the LoRA adapters.

## 5. Conclusion

We introduce UniDFlow, a unified vision–language diffusion model that performs understanding, text-to-image generation, and instruction-guided editing within a single discrete flow-matching framework. We further propose mRef-DPO, a reference-guided multimodal preference objective that jointly aligns text and image outputs relative to a frozen reference policy, improving faithfulness and controllability. Extensive results across six benchmarks show consistent gains, underscoring modality-aware preference alignment as critical for robust reasoning-grounded generation and precise visual editing.

## Impact Statement

This work presents a unified multimodal generative system that combines high-level understanding with high-fidelity visual generation. Such systems can enhance accessibility, creativity, and productivity by enabling natural multimodal interaction, supporting educational and design workflows, and improving human–computer interfaces. Our parameter-efficient training approach can also reduce computational cost compared to large-scale end-to-end retraining, potentially lowering environmental impact.

At the same time, improved generation and editing capabilities introduce risks. High-quality multimodal synthesis can be misused for deceptive media manipulation and precise editing may enable subtle alterations that are difficult to detect. Biases in pretrained vision–language backbones may also propagate into generated outputs, leading to stereotypical or harmful representations.

Our reference-based multimodal preference alignment in stage III aims to improve faithfulness and controllability by learning relative preferences under shared conditioning. This can help reduce spurious correlations and limit the amplification of dataset-specific artifacts when preference data is more balanced. However, alignment quality depends on the diversity and representativeness of the supervision signals, and misuse risks remain. Responsible deployment should therefore include safeguards such as content moderation, bias evaluation, and transparency mechanisms (*e.g.*, watermarking or provenance tracking).

## Acknowledgments

This research was partially supported by Google, the Google TPU Research Cloud (TRC) program, the National Science Foundation CAREER Award #2542328, the U.S. Defense Advanced Research Projects Agency (DARPA) under award HR001125C0303, and the U.S. Army under contract W5170125CA160. The views and conclusions contained herein are those of the authors and should not be interpreted as necessarily representing the official policies, either expressed or implied, of Google, NSF, DARPA, the U.S. Army, or the U.S. Government. The U.S. Government is authorized to reproduce and distribute reprints for governmental purposes notwithstanding any copyright annotation therein.

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

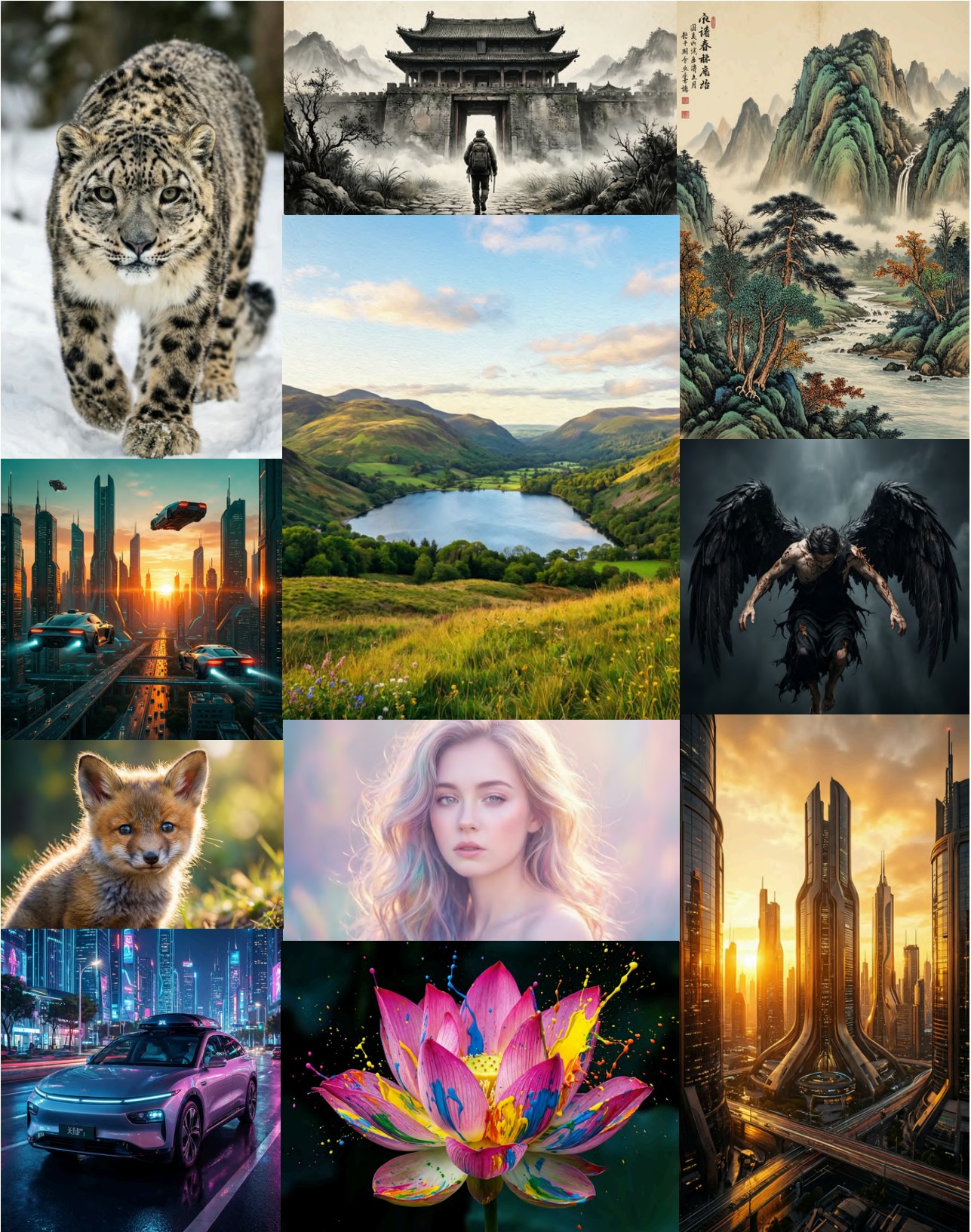

*Figure 9.* Image Generation with UniDFlow

# A. Implementation Details

We employ a three-stage training pipeline (Table 7) that progressively builds (i) visual instruction-following capability, (ii) high-fidelity visual generation, and (iii) joint multimodal understanding and alignment. Across all stages, we use AdamW optimization with mixed-precision training and gradient clipping to stabilize training at scale.

**Stage I: Text Alignment.** We first perform supervised fine-tuning to teach the model to follow visual instructions and ground text responses in images. To improve robustness to real-world inputs, we train with variable aspect ratios and variable image resolutions, enabling the model to generalize across diverse image formats. The learning-rate schedule uses a warmup phase followed by cosine annealing for stable convergence. Additionally, we set $\lambda_{\mathrm{KL}} = 1.8$.

**Stage II: Visual Alignment.** Next, we train the model for visual generation using a diffusion-based objective. We train at multiple resolutions (with variable aspect ratios) to encourage both global structure and fine detail, and use a linear learning-rate schedule with a longer warmup to support stable optimization under the generative objective. Regularization is applied via weight decay to improve generalization in this stage.

**Stage III: Reference-Based Multimodal Preference Alignment.** Finally, we jointly optimize understanding and alignment, combining multimodal comprehension with aligned outputs. We expand the image-resolution range further and increase the maximum sequence length to support longer-context reasoning over visual content. This stage uses a cosine-annealed schedule with warmup and moderate regularization, aiming to consolidate gains from the first two stages while maintaining training stability at scale.

**Throughput–size trade-off.** Figure 10 summarizes the empirical efficiency landscape by plotting inference throughput against model size for a set of representative systems (Janus-Pro (Chen et al., 2025a), OmniGenV2 (Wu et al., 2025b), BAGEL (Deng et al., 2025), MammothModa2 (Shen et al., 2025), EMMA (He et al., 2025), and MUDDIT (Shi et al., 2025)) and our variants at 0.7B, 4B, 8B, and 14B parameters. Rather than exhibiting a strictly monotonic dependence on parameter count, the scatter shows substantial dispersion across independently implemented models, indicating that architectural choices and inference stacks materially affect end-to-end throughput beyond raw scale. In the large-model regime (∼14B), UniDFlow-14B attains the highest throughput among the compared methods, outperforming other models of similar size (*e.g.*, BAGEL

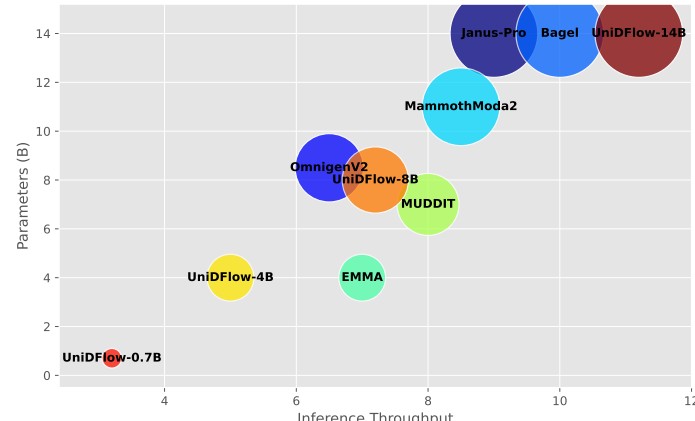

*Figure 10.* Inference throughput versus parameter count (in billions) for representative baselines and our model family. Higher throughput (right) is better, while fewer parameters (down) are more compact.

and Janus-Pro), suggesting improved runtime efficiency at scale. At intermediate sizes (∼7–9B), UniDFlow-8B is competitive with contemporaneous baselines, while the smaller UniDFlow-0.7B and UniDFlow-4B provide lightweight operating points that prioritize compactness with correspondingly lower throughput.

# B. Training Data

We employ a three-stage data curriculum that progressively transitions from supervised multimodal instruction learning to large-scale image-text pretraining and finally preference-based alignment for unified understanding, generation, and editing. Table 6 summarizes the training data.

**Token accounting.** Throughout this work, the reported token counts include *both* text tokens and discretized/embedded image tokens as consumed by the multimodal sequence interface (*i.e.*, the effective sequence length seen by the transformer). We report aggregate tokens per stage.

**Stage I: Text Alignment.** In Stage 1, we initialize instruction-following behavior using MMInstruct (Liu et al., 2024b), a high-quality multimodal instruction tuning dataset spanning diverse domains and instruction types. We use approximately 1.0M image–prompt–answer examples (MMInstruct reports 973K instructions (Liu et al., 2024b)) and train for ≈ 0.6T total (image+text) tokens.

*Table 7.* Training setup by stage. Stages I–III correspond to instruction tuning, visual generation, and joint understanding/alignment.

| Hyperparameter / Setting | Stage I | Stage II | Stage III |
|---|---|---|---|
| GPUs | 32×A100 (80GB) | 32×H100 (80GB) | 48×H100 (80GB) |
| Batch / GPU | 8 | 6 | 4 |
| Optimizer | AdamW | AdamW | AdamW |
| Init LR | $1\times10^{-5}$ | $5\times10^{-5}$ | $2\times10^{-5}$ |
| LR schedule | Cosine | Linear | Cosine |
| Warmup steps | 200 | 1000 | 1200 |
| Train steps | 10K | 25K | 30K |
| Grad accumulation | 6 | 4 | 4 |
| Max grad norm | 2.0 | 1.0 | 2.0 |
| Weight decay | 0 | $1\times10^{-2}$ | $1\times10^{-3}$ |
| Diffusion steps | 40 | 50 | 50 |
| Classifier-free guidance | 8 | 8 | 12 |
| Resolution | 224–1024 | 256/512/768/1024 | 224–1280 |
| Aspect ratios | 1:1, 4:3, 3:4 | 1:1, 16:9, 9:16 | 1:1, 4:3, 3:4, 16:9, 9:16 |
| Max seq length | 2048 | 2048 | 4096 |
| Precision | BF16 | BF16 | BF16 |
| GPU-Hours | 256 | 320 | 528 |

**Stage II: Visual Alignment.** Stage 2 focuses on *text-to-image generative pretraining* to improve prompt adherence, compositional generalization, and broad visual coverage. We sample a total of $\approx 4.5$M images (with associated text prompts/captions) from three sources: (i) 1.5M from LAION-5B (Schuhmann et al., 2022), (ii) 1.0M from JourneyDB (Sun et al., 2023), and (iii) 2.0M from the `jackyhate/text-to-image-2M` collection on Hugging Face (Jackyhate, 2024). We refine and normalize the paired text using a proprietary LLM-based caption/prompt rewriting pipeline to reduce noise and increase instruction clarity, and train for $\approx 1.2$T total (image+text) tokens.

*Table 6.* Stage-wise data summary (image+text tokens).

| St. | Obj. | | # Tok. |
|---|---|---|---|
| 1 | SFT (MMInstruct) | 1.0M | 0.6T |
| 2 | T2I (refined) | 4.5M | 1.2T |
| 3 | Pref align. | 3.5M | 1.8T |
| **Total** | | | 3.6T |

**Stage III: Reference-Based Multimodal Preference Alignment.** Stage 3 aligns the model to high-quality, instruction-faithful outputs in our unified data format for (a) multimodal understanding, (b) image generation, and (c) image editing. We curate $\approx 3.5$M base tasks by aggregating: OpenGPT-4o-Image ($\sim$80K) (Chen et al., 2025b), AnyEdit-derived edits ($\sim$3.0M; AnyEdit reports 2.5M editing pairs) (Yu et al., 2024b), and Pico-Banana-400K ($\sim$400K) (Qian et al., 2025). We then convert these tasks into a high-quality preference dataset via rejection-sampling style annotation using proprietary multimodal LLMs: for each edit instance, we generate and store (i) *reflection* traces with a positive:negative ratio of $3:6$, and (ii) paired instruction/response candidates with positive:negative ratio $4:10$ (stored as accepted vs. rejected candidates in our format). Stage 3 consumes $\approx 1.8$T total tokens.

Table 8 reports accuracy on eight visual reasoning subtasks, Recognition, Referring, Spotting, Extraction, Parsing, Calculation, Understanding, and Reasoning, together with their macro Average on OCRBenchV2 (Fu et al., 2024). The compared systems include strong understanding-focused VLMs (*e.g.*, Qwen2.5-VL (Bai et al., 2025a), InternVL (Chen et al., 2024b)), unified understanding–generation models (*e.g.*, EMMA (He et al., 2025), BAGEL (Deng et al., 2025), MUDDIT (Shi et al., 2025), MammothModa2 (Shen et al., 2025)), and proprietary multimodal assistants (*e.g.*, GPT-4o, Gemini-Pro (Team et al., 2023)). Fig. 22 qualitatively shows UniDFlow's robustness on complex understanding. This evaluation is particularly diagnostic because it separates perceptual grounding (Recognition/Spotting/Extraction), structured interpretation (Parsing/Calculation), and holistic inference (Understanding/Reasoning).

Across model scales, our unified model family consistently dominates the subtask profile, with performance improving monotonically from Ours-4B → Ours-8B → Ours-14B. Concretely, Ours-14B achieves the best overall Average = 63.8, improving over the strongest baseline (MammothModa2, 56.1) by +7.7 points. Gains are broad rather than concentrated in a single capability: relative to the best previous work in every column, Ours-14B improves Recognition (76.7; +7.9), Referring (47.1; +7.6), Spotting (16.5; +2.8), Extraction (96.9; +4.3), Parsing (48.4; +9.2), Calculation (58.1; +7.9), Understanding (88.7; +8.5), and Reasoning (77.1; +9.0). Notably, the largest deltas occur in Parsing and Reasoning, suggesting that the proposed approach strengthens compositional/structured visual reasoning beyond raw perception.

*Table 8.* Evaluation of existing VLMs and MLLMs on English tasks of OCRBench v2 (Fu et al., 2024) public data. "Recognition", "Referring", "Spotting", "Extraction", "Parsing", "Calculation", "Understanding", and "Reasoning" refer to text recognition, text referring, text spotting, relation extraction, element parsing, mathematical calculation, visual text understanding, and knowledge reasoning, respectively. Higher values indicate better performance.

| Method | Recog. | Ref. | Spot. | Extr. | Pars. | Calc. | Und. | Reas. | Avg. |
|---|---|---|---|---|---|---|---|---|---|
| Qwen2.5-VL-7B (Bai et al., 2025a) | 68.8 | 25.7 | 1.2 | 80.2 | 30.4 | 38.2 | 73.2 | 56.2 | 46.7 |
| InternVL3-14B (Chen et al., 2024b) | 67.3 | 36.9 | 11.2 | 89.0 | 38.4 | 38.4 | 79.2 | 60.5 | 52.6 |
| GPT-4o (OpenAI, 2024) | 61.2 | 26.7 | 0.0 | 77.5 | 36.3 | 43.4 | 71.1 | 55.5 | 46.5 |
| GPT-4o-mini (OpenAI, 2024) | 57.9 | 23.3 | 0.6 | 70.8 | 31.5 | 38.8 | 65.9 | 55.1 | 43.0 |
| Gemini-pro (Team et al., 2023) | 61.2 | 39.5 | 13.5 | 79.3 | 39.2 | 47.7 | 75.5 | 59.3 | 51.9 |
| Qwen3-VL-8B (Bai et al., 2025b) | 64.4 | 38.2 | 5.7 | 91.03 | 37.8 | 44.2 | 76.8 | 62.6 | 55.7 |
| OmniGenV2 (Wu et al., 2025b) | 61.3 | 36.5 | 2.4 | 87.23 | 33.4 | 40.7 | 72.7 | 65.7 | 48.7 |
| BAGEL (Deng et al., 2025) | 65.8 | 37.1 | 3.3 | 90.45 | 38.5 | 41.3 | 75.2 | 66.4 | 52.2 |
| Emma (He et al., 2025) | 66.7 | 36.5 | 6.7 | 91.3 | 37.5 | 44.5 | 76.7 | 67.2 | 53.8 |
| MUDDIT (Shi et al., 2025) | 64.9 | 38.4 | 13.7 | 92.6 | 34.5 | 49.4 | 78.3 | 66.1 | 54.7 |
| MammothModa2 (Shen et al., 2025) | 68.2 | 39.5 | 11.4 | 92.2 | 39.1 | 50.2 | 80.2 | 68.1 | 56.1 |
| **UniDFlow-4B** | 69.9 | 41.2 | 12.9 | 94.1 | 42.2 | 53.4 | 83.1 | 70.8 | 58.4 |
| **UniDFlow-8B** | 72.2 | 43.8 | 14.9 | 95.0 | 45.7 | 55.1 | 85.9 | 73.5 | 60.7 |
| **UniDFlow-14B** | **76.7** | **47.1** | **16.5** | **96.9** | **48.4** | **58.1** | **88.7** | **77.1** | **63.8** |

A second takeaway is that even the compact variant (Ours-4B) is competitive with or better than substantially larger baselines: it reaches 58.4 Avg., exceeding MammothModa2 (56.1) and MUDDIT (54.7) while also improving the hardest "reasoning-heavy" columns (Calc. = 53.4, Reas. = 70.8). This aligns with the paper's core design choice: rather than entangling understanding and generation in shared parameters, the method trains separate lightweight adapters for understanding vs. generation and combines them with a learned router, reducing objective interference and preserving specialization.

## C. Full Quantitative Results on GenEval and DPGBench

The main paper reports the overall performance of UniDFlow on GenEval and DPGBench. Here, we provide the complete attribute-wise breakdown used by both benchmarks. Table 9 shows that UniDFlow achieves the best *global* score and consistently improves across fine-grained categories (*e.g.*, entity, attribute, relation) as well as compositional criteria (single/two-object, counting, color, position, and color-attribute). These results indicate that the gains are not driven by a single subset of prompts; instead, UniDFlow improves performance uniformly across evaluation dimensions, reflecting stronger text–image alignment and more reliable adherence to structured constraints.

## D. Additional Results

**Ablation on training tokens and LoRA rank.** We study the effect of (i) the total number of pre-training tokens (image+text) and (ii) the LoRA rank used for adaptation. Figure 13 shows a consistent improvement as we scale the training budget from 0.5T to 3T tokens, yielding substantial gains across *TextGen*, *GenEval* (Ghosh et al., 2023), *DPGBench* (Hu et al., 2024), and *ImgEdit-Bench* (Ye et al., 2025). We also ablate the LoRA rank and observe that increasing the rank from 8 to 32 produces the largest marginal improvement across all benchmarks, indicating that low ranks under-parameterize the adaptation. Beyond rank 32, performance improvements diminish and largely saturate up to rank 128, suggesting the adaptation becomes capacity-sufficient. We use a default LoRA rank of 64 in all experiments.

**Ablation on Stage-III losses.** Fig. 14 compares the visual results for image editing without alignment losses. Removing $\mathcal{L}_{\text{vRef-DPO}}$ weakens visual grounding. The model often preserves the coarse scene but fails to faithfully realize the requested edit, such as retaining the bird or producing incomplete candle lighting. Removing $\mathcal{L}_{\text{tRef-DPO}}$ degrades instruction following, leading to missing or semantically incorrect target objects, *e.g.*, the kitten is not properly placed on the leaf sailboat and the rabbit replacement becomes unreliable. Removing $\mathcal{L}_{\text{mRef-DPO}}$ introduces stronger cross-modal inconsistencies and visual artifacts, such as unrealistic object appearance, poor scene integration, or excessive color changes. In contrast, the full UniDFlow model better preserves the input context while executing the requested edit, producing a kitten plausibly navigating a leaf sailboat, replacing the bird with a scene-consistent rabbit, and lighting the candles without distorting the subject. These results show that the three alignment losses are complementary, with visual-reference alignment improving preservation and grounding, text-reference alignment improving semantic edit fidelity, and multimodal reference alignment

*Table 9.* Detailed text-to-image generation evaluation on DPG-BENCH and GENEVAL.

| DPG-BENCH | | | | | | |
|---|---|---|---|---|---|---|
| Model | Params | Global | Entity | Attribute | Relation | Other |
| DALL-E 3 (Betker et al., 2023) | – | 90.97 | 89.61 | 89.39 | 90.58 | 89.83 |
| SD3-Medium (Esser et al., 2024) | 2B | 87.92 | 91.01 | 88.48 | 80.72 | 86.81 |
| Qwen-Image (Wu et al., 2025a) | 7B+20B | 91.32 | 91.56 | 92.02 | 94.31 | 92.73 |
| TokenFlow-XL (Geyer et al., 2023) | 14B | 87.33 | 88.54 | 89.01 | 85.09 | 86.55 |
| Janus-Pro-7B (Chen et al., 2025a) | 7B | 86.91 | 88.95 | 89.43 | 90.02 | 89.48 |
| BAGEL (Deng et al., 2025) | 7B+7B | 89.42 | 91.43 | 90.42 | 92.34 | 88.78 |
| OmniGenV2 (Wu et al., 2025b) | 3B+4B | – | – | 86.43 | 91.23 | – |
| MammothModa-2 (Shen et al., 2025) | 8B+3B+2B | 81.16 | 92.99 | 90.16 | 94.35 | 84.81 |
| EMMA (He et al., 2025) | 4B | 91.24 | 91.71 | 90.59 | 92.23 | 90.02 |
| MUDDIT (Shi et al., 2025) | 7B | 89.42 | 90.47 | 89.56 | 90.72 | 88.63 |
| **UniDFlow** | 4B | **93.42** | **94.44** | **95.34** | **95.03** | **93.86** |

| GENEVAL | | | | | | | |
|---|---|---|---|---|---|---|---|
| Model | Params | Single Obj | Two Obj | Counting | Colors | Position | Color Attr |
| DALL-E 3 (Betker et al., 2023) | – | 0.96 | 0.87 | 0.47 | 0.83 | 0.43 | 0.45 |
| SD3-Medium (Esser et al., 2024) | 2B | 0.99 | 0.94 | 0.72 | 0.89 | 0.33 | 0.60 |
| Qwen-Image (Wu et al., 2025a) | 7B+20B | **1.00** | 0.95 | **0.93** | 0.92 | 0.87 | 0.83 |
| TokenFlow-XL (Geyer et al., 2023) | 14B | 0.95 | 0.60 | 0.41 | 0.81 | 0.16 | 0.24 |
| Janus-Pro-7B (Chen et al., 2025a) | 7B | 0.99 | 0.89 | 0.59 | 0.90 | 0.79 | 0.66 |
| BAGEL (Deng et al., 2025) | 7B+7B | 0.98 | 0.95 | 0.84 | 0.95 | 0.78 | 0.77 |
| OmniGenV2 (Wu et al., 2025b) | 3B+4B | 0.95 | 0.93 | 0.64 | 0.81 | 0.73 | 0.74 |
| MammothModa-2 (Shen et al., 2025) | 8B+3B+2B | **1.00** | 0.97 | 0.63 | 0.89 | **0.90** | 0.82 |
| EMMA (He et al., 2025) | 4B | **1.00** | **0.99** | 0.87 | **0.98** | 0.86 | **0.87** |
| MUDDIT (Shi et al., 2025) | 7B | 0.95 | 0.93 | 0.85 | 0.96 | 0.82 | 0.84 |
| **UniDFlow** | 4B | **1.00** | **0.99** | 0.89 | **0.98** | 0.87 | 0.83 |

stabilizing cross-modal credit assignment for more reliable reasoning-grounded generation.

# E. Extended Related Work

**Diffusion for Visual Generation.** Diffusion probabilistic models (DPMs) (Ho et al., 2020; Nichol et al., 2021; Saharia et al., 2022) surpass GANs (Goodfellow et al., 2014) in stability and generation quality, but are computationally expensive due to pixel-space diffusion. Latent diffusion models (LDMs) (Rombach et al., 2022) mitigate this cost by operating in a compressed latent space and achieve strong text-to-image performance (Zhang et al., 2023; Chen et al., 2024a; Podell et al., 2023). However, continuous Gaussian diffusion is well-suited for images but less natural for discrete modalities such as text. Discrete diffusion (Austin et al., 2021) addresses this by using categorical corruption (*e.g.*, masking), motivating image generators that replace autoregressive decoding with parallel mask-and-predict refinement, improving both fidelity and latency (Gu et al., 2022; Chang et al., 2022).

**LLMs and VLMs for Understanding.** Large language models (LLMs) (Touvron et al., 2023; Liu et al., 2024a) have achieved strong zero-shot reasoning and instruction-following by autoregressively generating tokens with decoder-only Transformers (Vaswani et al., 2017). Inspired by their success, vision–language models (VLMs) (Bai et al., 2023; 2025a) extend LLMs to visual inputs by coupling a vision encoder (*e.g.*, SigLIP (Zhai et al., 2023)) with a language model via lightweight projection layers, treating images as sequences of visual tokens. Models such as Qwen (Bai et al., 2025b), LLaVA (Liu et al., 2023), BLIP-2 (Li et al., 2023), and Flamingo (Alayrac et al., 2022) enable strong visual understanding (*e.g.*, captioning, VQA), but treat vision as read-only and rely on separate diffusion models for image generation.

Beyond likelihood training, preference alignment has been extended from LLMs to diffusion models using DPO-style objectives (Rafailov et al., 2023). Diffusion-DPO (Wallace et al., 2024) directly fine-tunes text-to-image models on pairwise human judgments via a likelihood-based preference loss, while DSPO (Zhu et al., 2025) instead aligns the diffusion score function in score space, staying closer to the original training objective. Subsequent variants such as DGPO (Luo et al., 2025) improve stability through group-wise preference optimization, and recent work further generalizes DPO-style alignment to discrete diffusion processes (Borso et al., 2025), bridging continuous and categorical diffusion frameworks.

**Unified Models for Understanding and Generation.** Diffusion models and vision–language models excel at generation and semantic understanding, respectively, motivating unified architectures. To improve generation and editing, recent models

add additional alignment stages. OmniGenV2 (Wu et al., 2025b) employs multimodal reflection for self-correction, while MammothModa2 (Shen et al., 2025) applies reinforcement learning with scalar rewards (*e.g.*, OCR and aesthetic scores). In contrast, UniDFlow introduces reference-based preference alignment across text and vision with reflection, enabling stable and faithful generation and editing.

---

**Dataset curation prompt used in our pipeline**

```
You are an expert data curator for multimodal image-editing instruction datasets.

Inputs:
1) SOURCE_IMAGE: the original image
2) RAW_INSTRUCTION: the original instruction text
3) EDITED_IMAGE: the ground-truth edited image

Tasks:
1) Produce clean training fields.
2) Create preference pairs by writing one "chosen" output and >=3 "rejected" alternatives (plausible but
worse).

Rules:
- Preserve intent; if ambiguous, pick the interpretation that matches EDITED_IMAGE.
- Be consistent with what is visible in SOURCE_IMAGE and EDITED_IMAGE; do not invent unseen details.
- Be explicit: what changes, where, how much/style constraints, and what must remain unchanged if relevant.
- Reflection describes the final edited image relative to the original; only observable outcomes.

Rejected alternatives (>=3):
- Each must be a realistic mistake (not nonsense) and include:
  - worse tuned instruction
  - worse edit directive
  - worse reflection of the incorrect outcome
  - "why_rejected" explaining what is wrong vs the chosen
  - "negative_type" from: under-edit, over-edit, wrong-attribute, wrong-region, unwanted-add/remove
- Include at least one near-miss (almost correct but subtly wrong: intensity, shade, lighting, missed
constraint).

Output JSON (return ONLY valid JSON; exactly these keys):
{
  "prompt": string,
  "image": "SOURCE_IMAGE",
  "answer_instruction_tuned": string,
  "edit_instruction_for_image": string,
  "edited_image": "EDITED_IMAGE",
  "reflection_of_edited_image": string,
  "preference_data": {
    "prompt": string,
    "chosen": {
      "answer_instruction_tuned": string,
      "edit_instruction_for_image": string,
      "reflection_of_edited_image": string
    },
    "rejected": [
      {
        "answer_instruction_tuned": string,
        "edit_instruction_for_image": string,
        "reflection_of_edited_image": string,
        "why_rejected": string,
        "negative_type": string
      }
    ]
  }
}

Now process:
SOURCE_IMAGE: <<IMAGE>>
RAW_INSTRUCTION: <<INSTRUCTION_TEXT>>
EDITED_IMAGE: <<EDITED_IMAGE>>
```

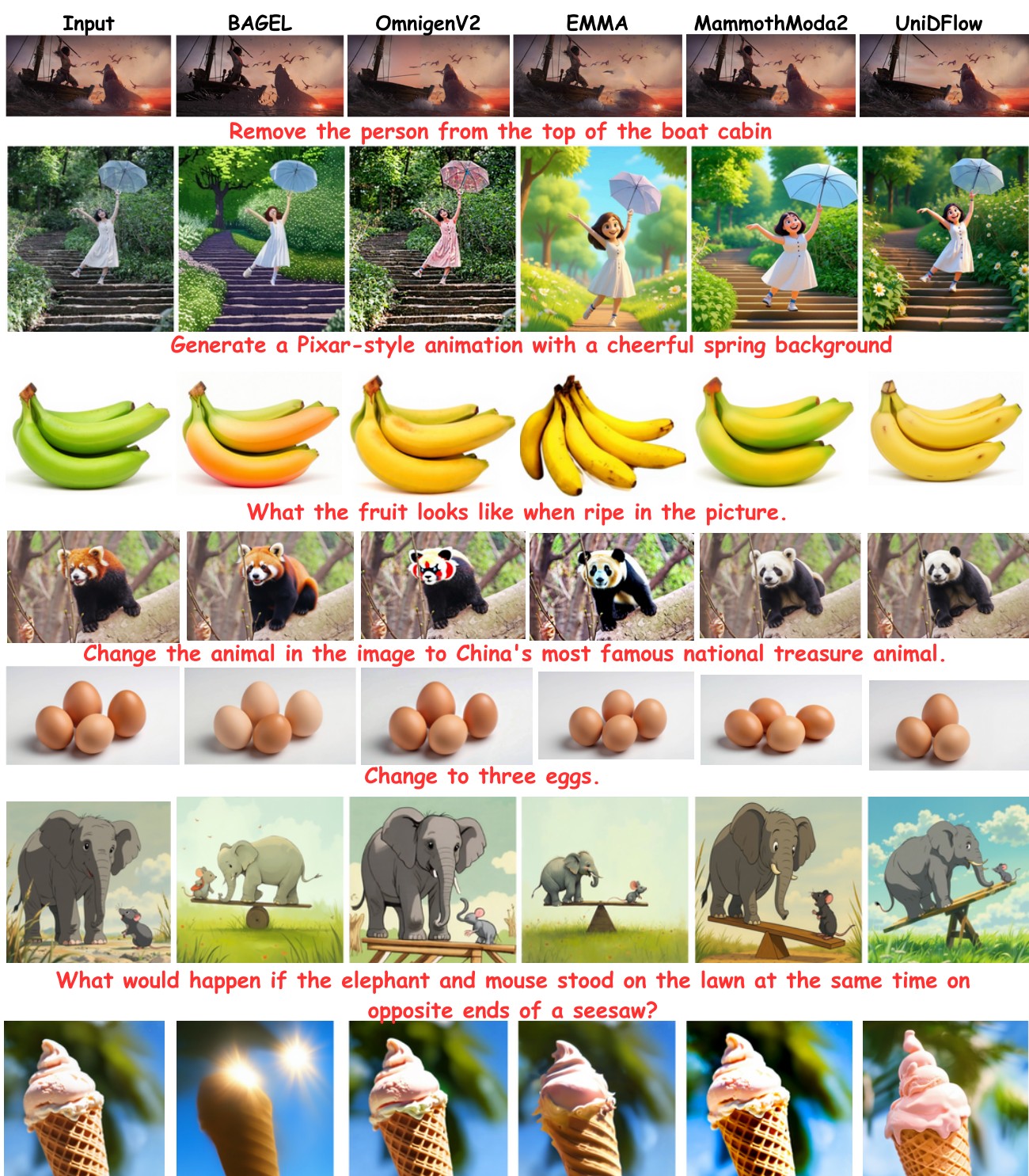

Figure 11. Qualitative comparison of reasoning-based image editing on complex images. Given an input image and a natural-language instruction, each column shows the result produced by a different editing method. The examples highlight challenging edits that require semantic understanding and commonsense reasoning, including object removal, style transformation, ripeness prediction, species replacement, quantity modification, physical interaction reasoning, and sunlight-induced melting effects.

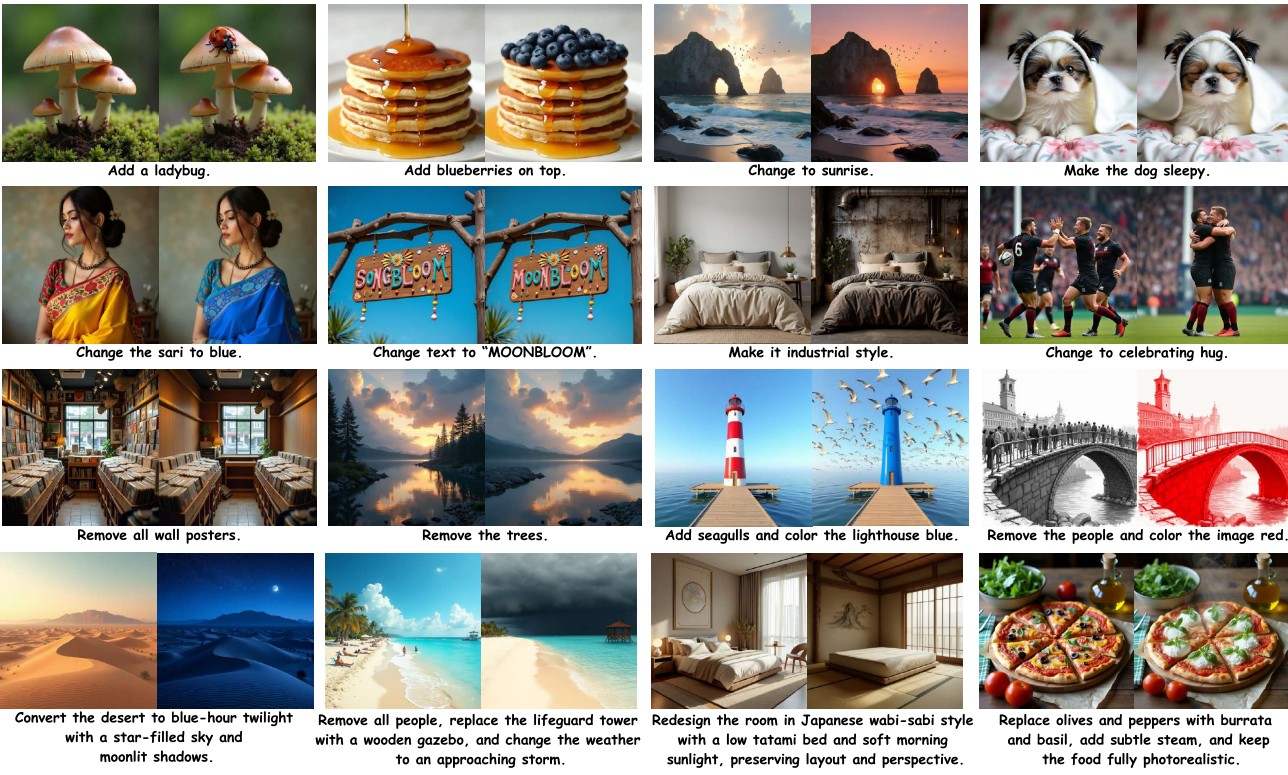

*Figure 12.* Image editing results on complex scenarios involving object, style, text, color, weather, and layout transformations.

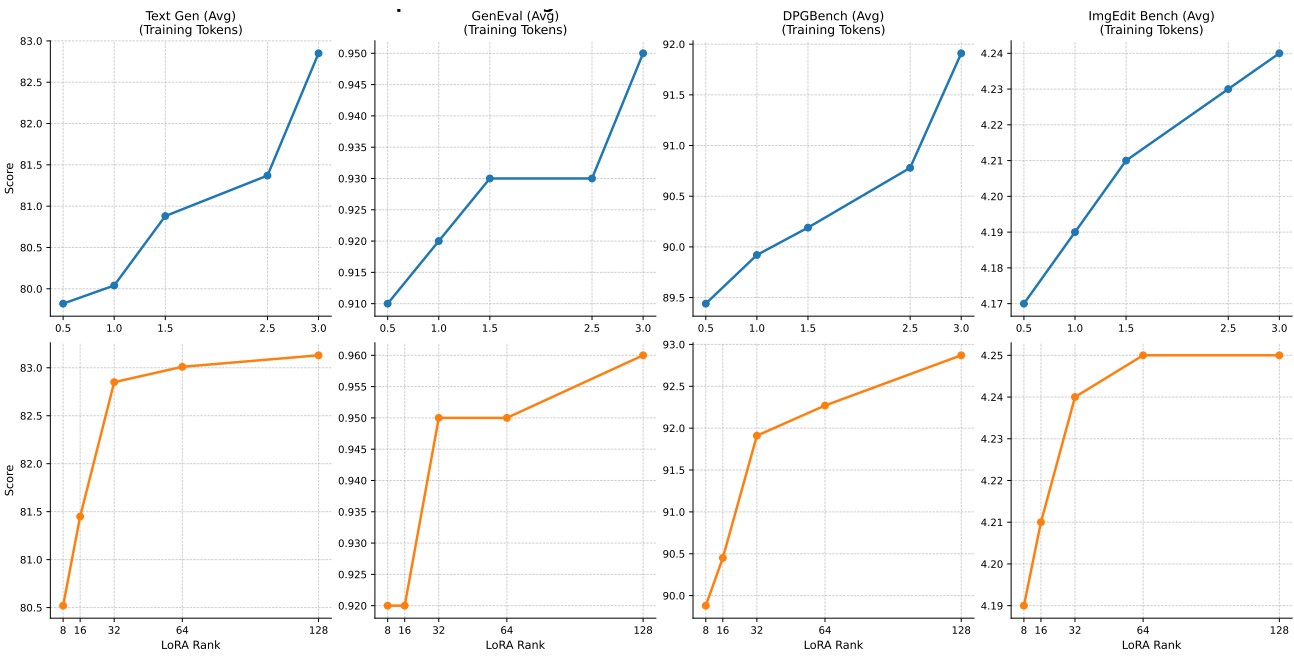

*Figure 13.* Analysis of the number of training tokens and LoRA ranks used across all stages.

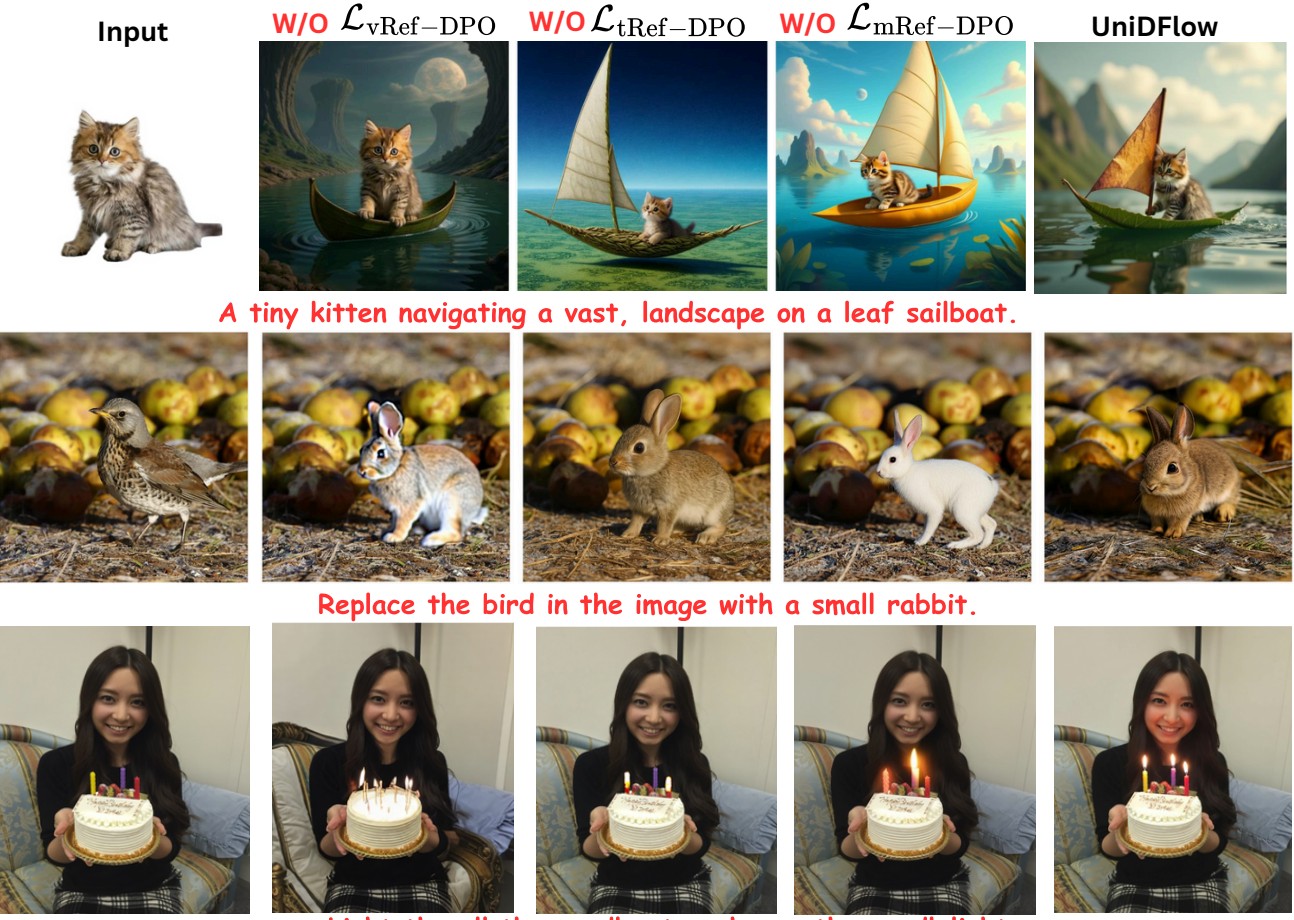

*Figure 14.* Visual comparison for Stage-III alignment losses.

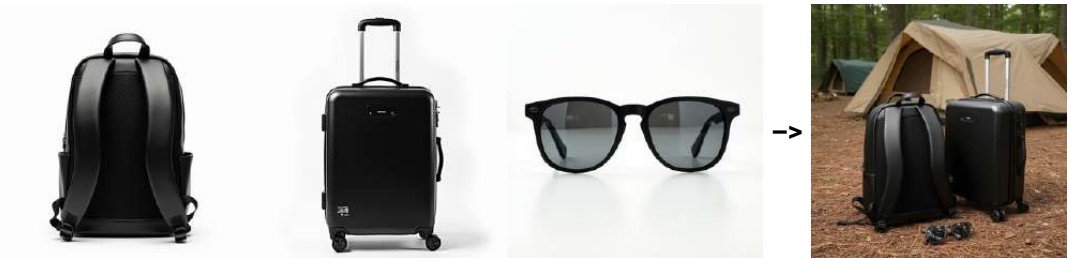

**A backpack, a suitcase, and sunglasses placed on the ground at a campsite.**

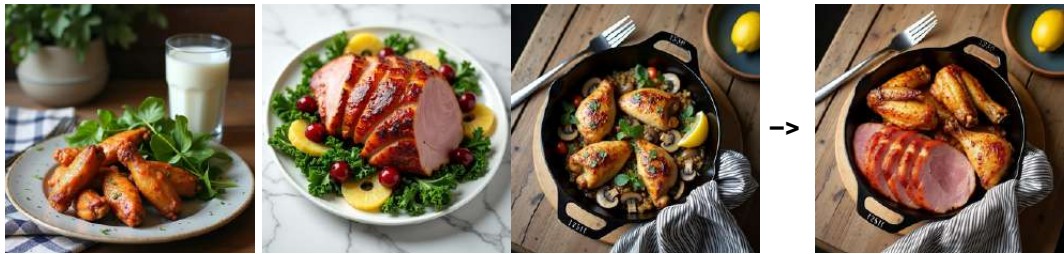

**A delicious meal featuring roasted chicken wings, sliced ham, and pan-seared chicken drumsticks served together on a wooden table.**

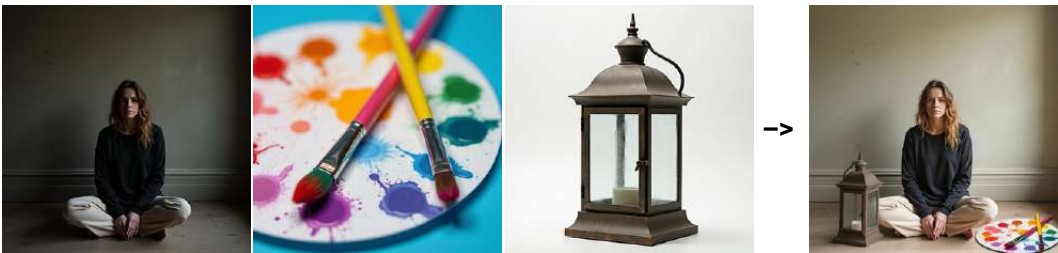

**A woman sitting indoors, with a vintage lantern and a painter's palette nearby, soft natural light.**

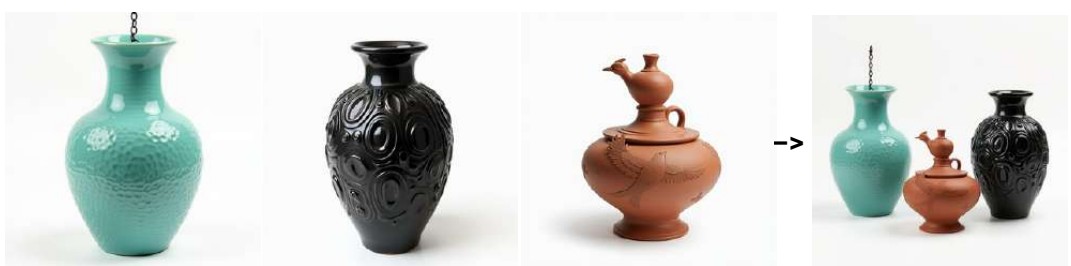

**Arrange all three reference vases in one image with consistent lighting and a balanced composition.**

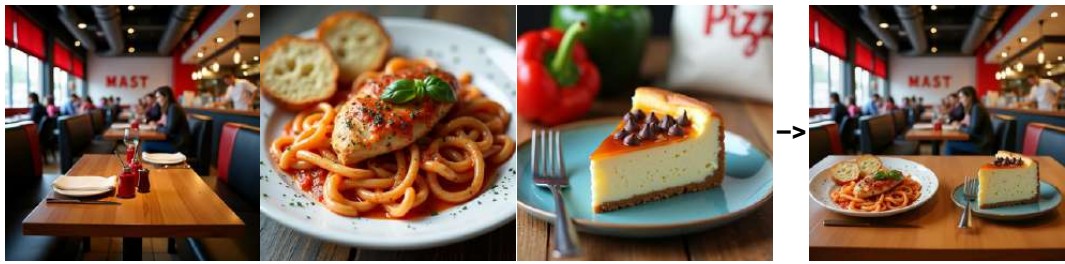

**Pasta, cheesecake, and a modern restaurant interior in one cohesive, realistic scene.**

*Figure 15.* Zero-shot multi-subject reasoning-based editing.

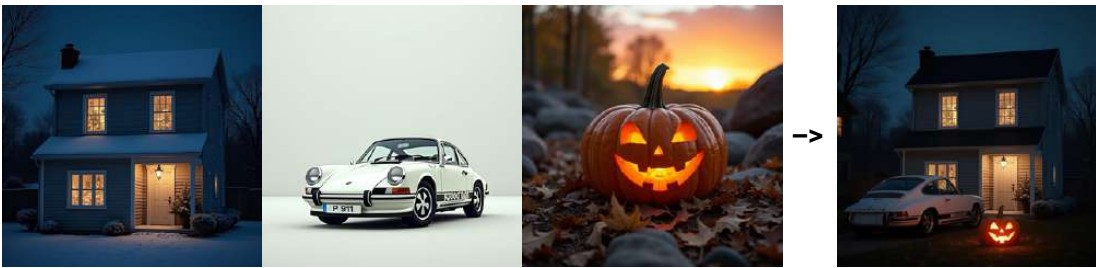

**A nighttime neighborhood scene featuring a cozy house, a vintage white car, and a carved pumpkin glowing at dusk.**

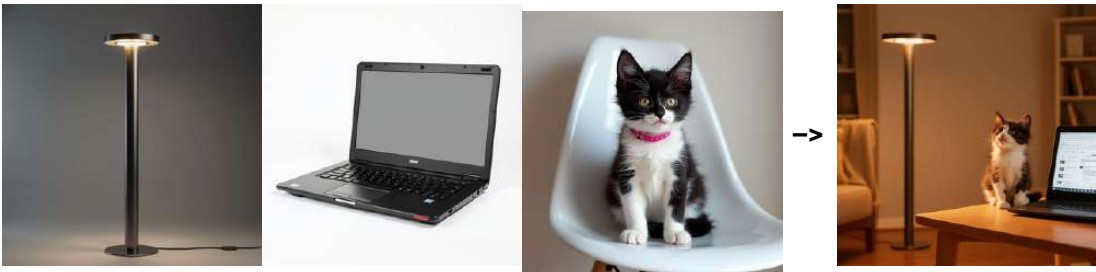

**A cozy indoor workspace with a sleek floor lamp, a laptop on the table, and a curious kitten sitting close by.**

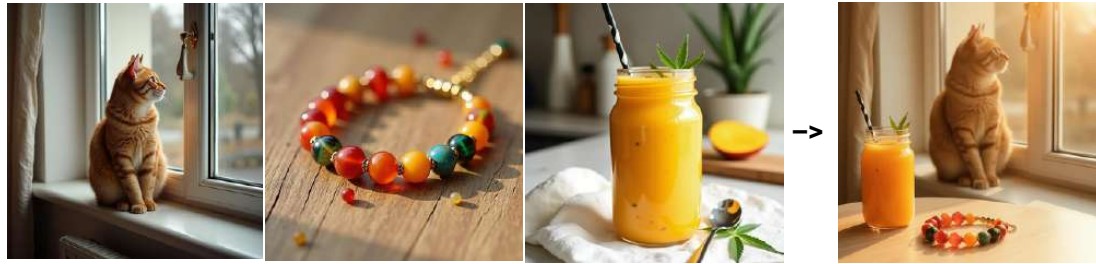

**A warm, sunlit interior with a cat sitting by the window, a jar of juice, and a beaded bracelet placed casually on a table.**

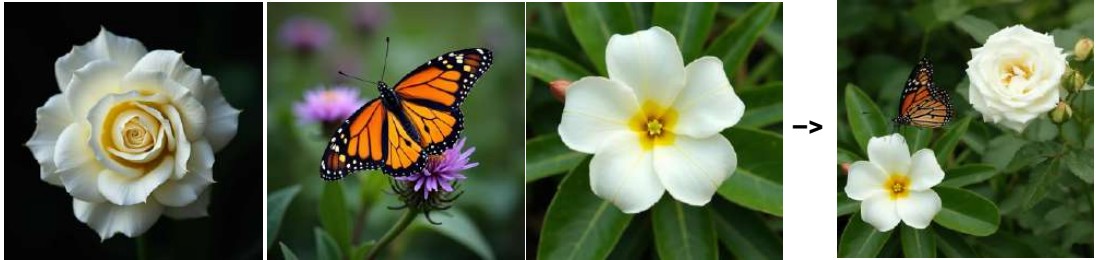

**A close-up nature scene featuring a white flower, a butterfly resting nearby, and a blooming white rose.**

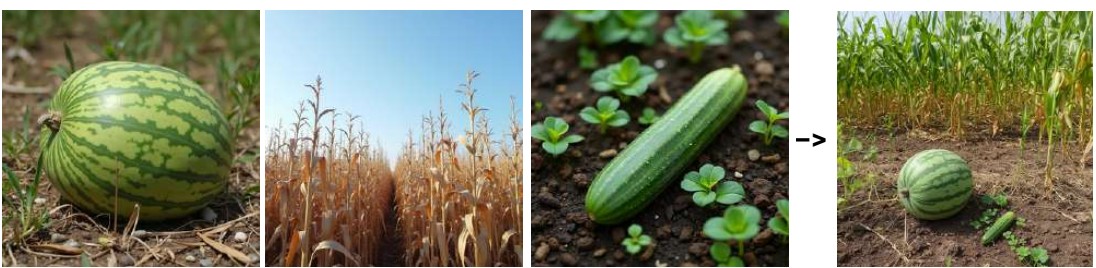

**A rural field scene with rows of corn, a watermelon resting on the soil, and a cucumber growing among young plants.**

*Figure 16.* Zero-shot multi-subject reasoning-based editing.

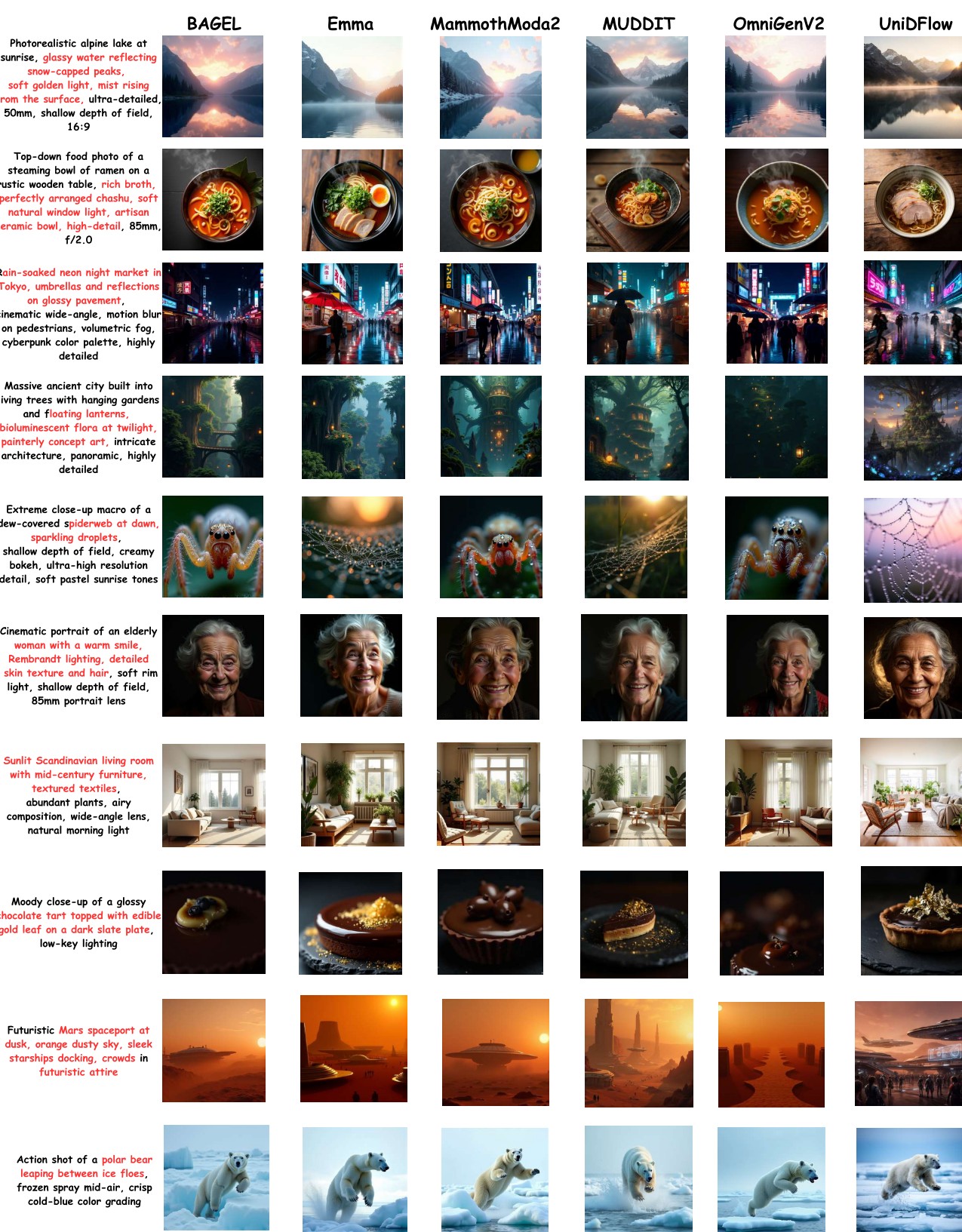

*Figure 17.* Text-to-image generation comparison against baselines.

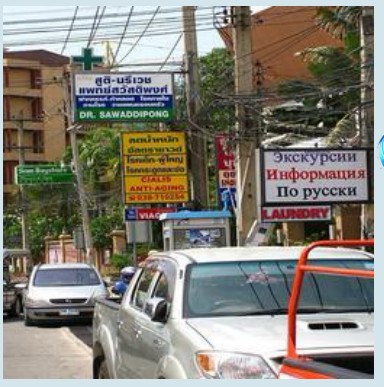

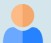 Describe the urban street scene shown in this image.

The image shows a busy, sunlit urban street with light traffic and parked vehicles, including several pickup trucks in the foreground. Low-rise buildings sit close to the road, partly shaded by leafy trees, while tall utility poles and a dense web of overhead power lines dominate the skyline. The streetscape is cluttered with prominent storefront and roadside signs in multiple languages, Thai and English on several boards

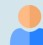 Describe the urban environment shown in this image.

The scene looks like a street-level corner of a city building with a large, grid-framed window revealing the inside of a casual fast-food/café space. A strip of signage along the bottom of the window advertises menu items (including "HOT WRAPS"), while a few patrons sit inside at tables. Outside, the building has a heavy stone/concrete facade with a small ventilation grate,

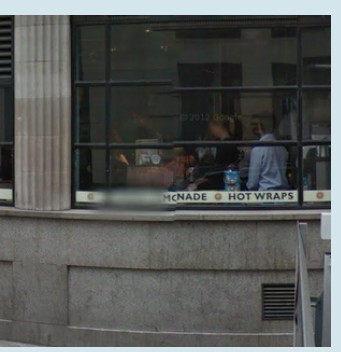

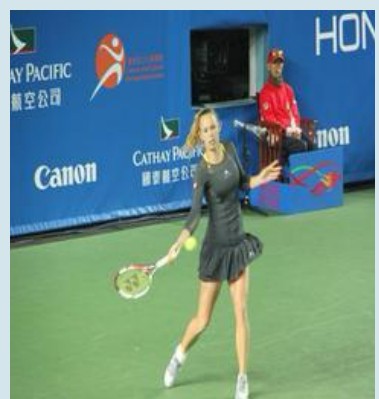

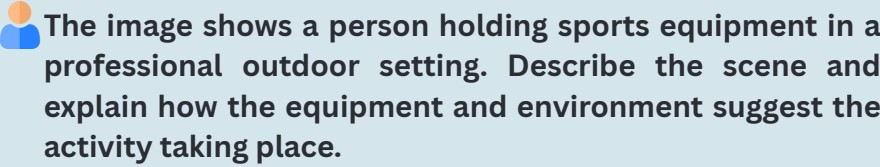 The image shows a person holding sports equipment in a professional outdoor setting. Describe the scene and explain how the equipment and environment suggest the activity taking place.

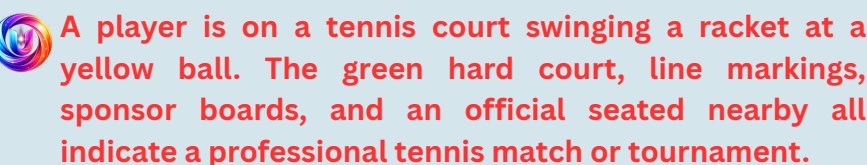 A player is on a tennis court swinging a racket at a yellow ball. The green hard court, line markings, sponsor boards, and an official seated nearby all indicate a professional tennis match or tournament.

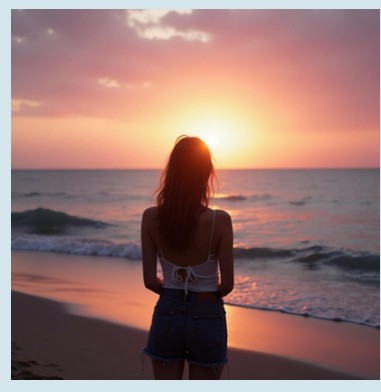

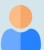 What does costal scene represents ?

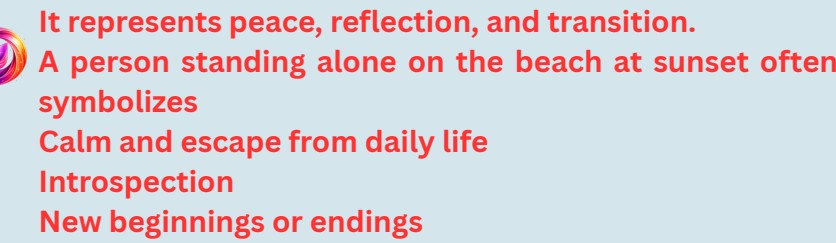 It represents peace, reflection, and transition.
A person standing alone on the beach at sunset often symbolizes
Calm and escape from daily life
Introspection
New beginnings or endings

*Figure 18.* Image-to-text generated results with UniDFlow.

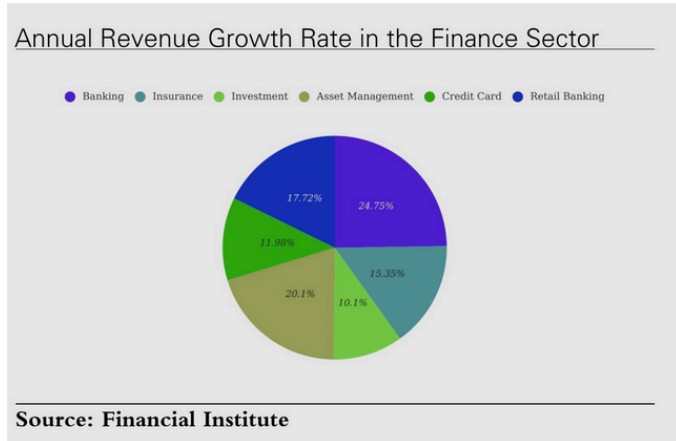

👤 **Convert the key information in the chart into a nested python dict.**

🌀 **"{\"title\": \"Annual Revenue Growth Rate in the Finance Sector\", \"source\": \"Source: Financial Institute\", \"x_title\": \"None\", \"y_title\": \"None\", \"values\": {\"Banking\": \"25.0%\", \"Insurance\": \"15.5%\", \"Investment\": \"10.2%\", \"Asset Management\": \"20.3%\", \"Credit Card\": \"12.1%\", \"Retail Banking\": \"17.9%\"}}"**

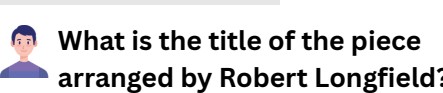
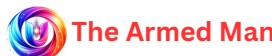

👤 **What is the title of the piece arranged by Robert Longfield?**

🌀 **The Armed Man**

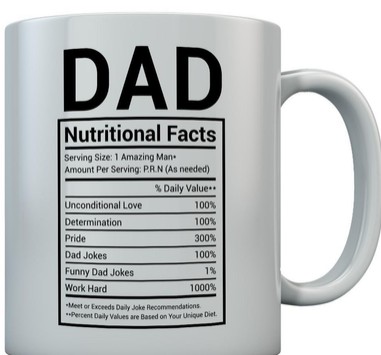

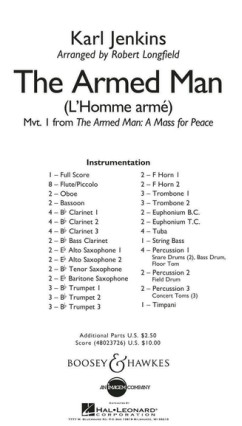

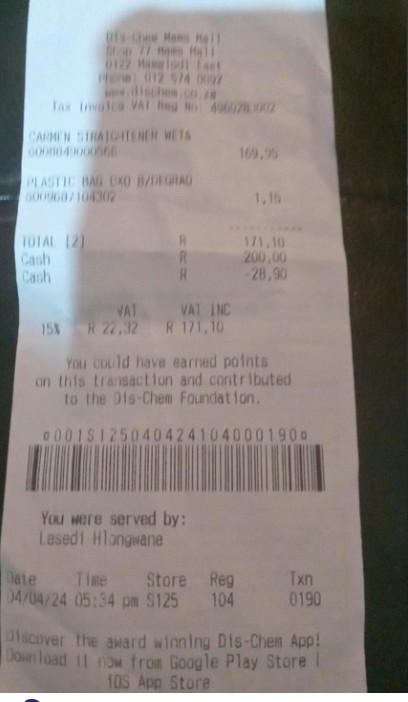

👤 **What is the serving size of the "nutrients" listed on the mug?**

🌀 **1 Amazing Man**

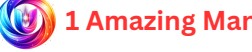

👤 **what brand of watch is this?**

🌀 **invicta**

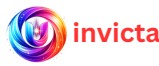

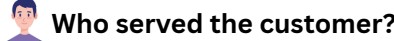

👤 **Who served the customer?**

🌀 **lesedi Hlongwane**

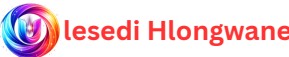
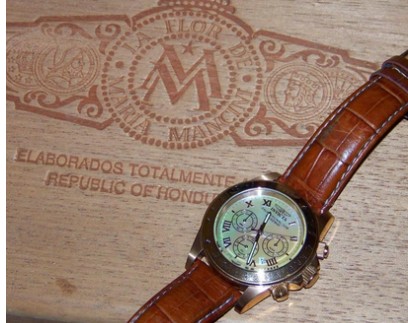

*Figure 19.* Image-to-text generated results with UniDFlow.

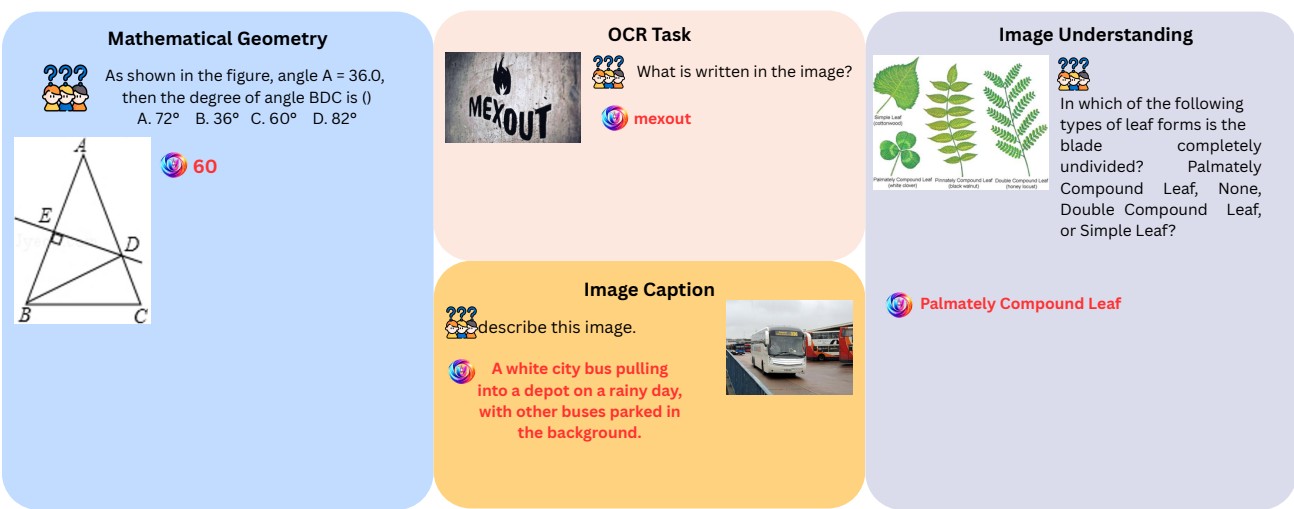

*Figure 20.* More complex reasoning tasks with UniDFlow.

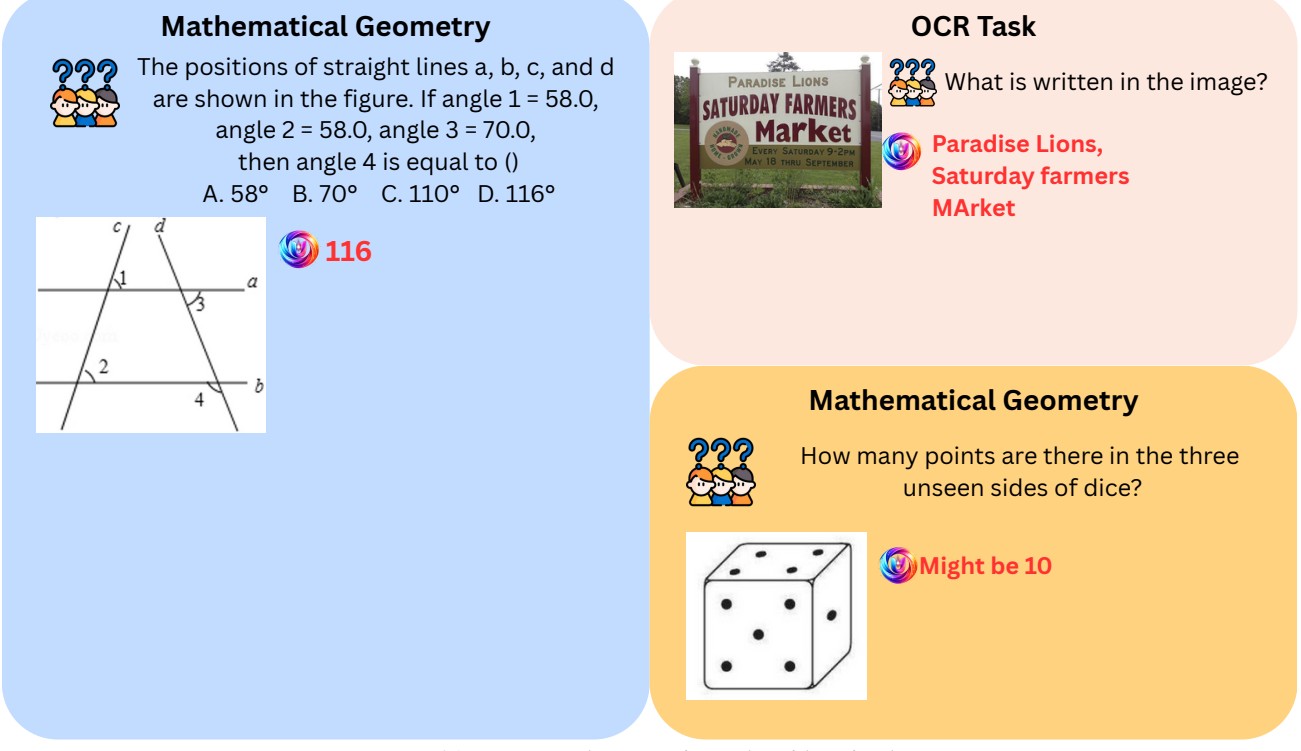

*Figure 21.* More complex reasoning tasks with UniDFlow.

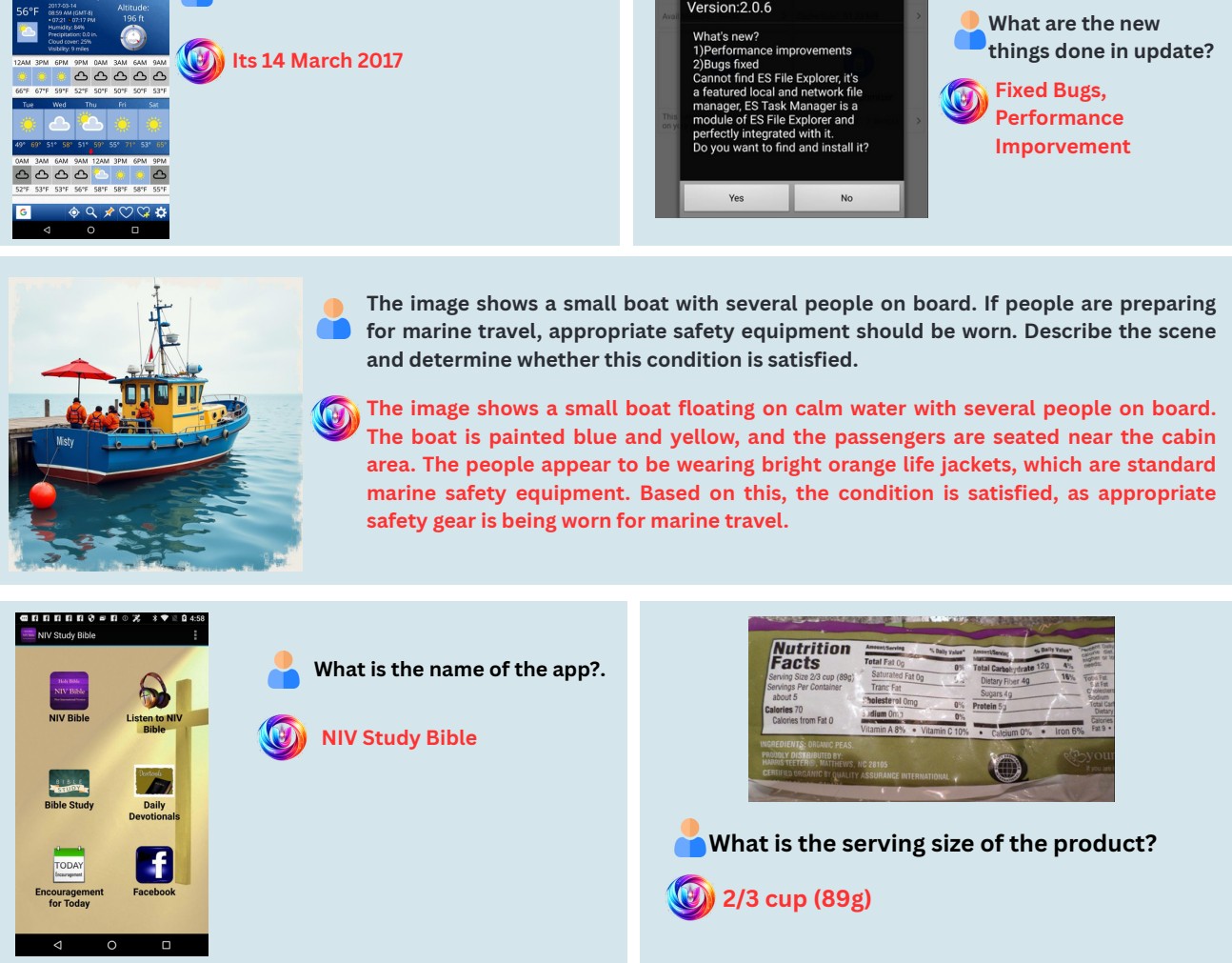

*Figure 22.* Image understanding and reasoning with complex scenes.

