# OpenReview forum: "Best of Both Worlds: Multimodal Reasoning and Generation via Unified Discrete Flow Matching"
_ICML.cc/2026/Conference — ICML 2026 regular_

### Official Review · Reviewer_vGVy · 2026-03-10

**Soundness:** 4
**Presentation:** 4
**Significance:** 3
**Originality:** 3
**Overall Recommendation:** 5
**Confidence:** 4

**Summary:**

The paper introduces UniDFlow, a unified framework for multimodal understanding and generation built upon a Vision-Language Model (VLM) backbone. UniDFlow first adapts the VLM to a discrete diffusion model using Time-Step Guided RMSNorm. It then leverages parameter isolation across three training stages to progressively equip the model with capabilities in understanding, generation, and preference alignment, all while preserving the strong priors of the pretrained VLM.

**Compliance With Llm Reviewing Policy:**

Affirmed.

**Final Justification:**

Thank you for the detailed rebuttal and the additional clarifications. My main concerns have been adequately addressed.

In particular, I appreciate the added explanation of why Time-Step Guided RMSNorm improves training stability, the denoising-step ablation clarifying the role of the 20-step setting, and the stronger discussion of how UniDFlow differs from and improves upon BAGEL. The clarification on scalability across multiple backbone sizes is also helpful and strengthens the practical value of the paper. The planned revisions to the figures and minor presentation issues further improve my confidence in the final version.

Overall, the rebuttal resolves my key questions, and I remain supportive of acceptance.

**Key Questions For Authors:**

Please refer to Weaknesses.

**Limitations:**

yes

**Strengths And Weaknesses:**

Strengths：
1.The writing is clear and logically structured.
2.The paper provides a thorough summary of the limitations in existing hybrid model architectures, and the proposed methods directly address those issues.
3.The experiments are rich in detail and the results are solid, with baselines that cover most state-of-the-art hybrid models, lending credibility to the findings.

Weaknesses：
1.There are several minor errors, including misplaced formulas and incorrect headers.
2.The paper lacks an analysis of why Time-Step Guided RMSNorm contributes to training stability, comparing to other methods such as using semantic features or hidden embeddings for diffusion models.
3.One of the contributions claimed in the introduction—“UniDFlow achieves efficient training and inference, requiring only 20 denoising steps while preserving high generation quality”—is not clearly linked to the proposed innovations, nor is it supported by dedicated experiments.
4.Figures 3 and 4 are somewhat confusing and make UniDFlow appear like a black box.
5.UniDFlow shares many similarities with BAGEL, but the ablation studies do not clarify which specific improvements lead to the performance gains shown in Table 1.
6.If possible, demonstrate that UniDFlow scales effectively to larger models, it would significantly enhance the value of the work.

---

> ### Author Rebuttal · Authors · 2026-03-30
>
> We thank the reviewer for the thoughtful and positive feedback. We are glad that the reviewer found the paper clearly written, the problem analysis well motivated, and the experimental results convincing.
>
> **Q1: Minor Typos**
>
> **Answer:** We thank the reviewer for the careful reading. We have identified the misplaced formulas and incorrect headers, and will thoroughly revise the manuscript in the final camera-ready version.
>
> **Q2: Analysis of TSGN**
>
> **Answer:** We thank the reviewer for this valuable suggestion. The main motivation for TSG-RMSNorm is to enable stable adaptation of a pretrained RMSNorm-based VLM to discrete diffusion. Instead of injecting timestep information directly into attention or MLP activations, which can perturb learned feature statistics, TSG-RMSNorm conditions time by modulating the pretrained RMSNorm parameters with zero-initialized, time-dependent scale and bias terms. This allows the model to exactly recover the pretrained backbone at initialization while introducing diffusion-time information through controlled rescaling rather than altering hidden-state geometry. For an ablation comparing TSG-RMSNorm with AdaLN and In-Context guidance of timesteps, please see our response to reviewer `ZmRr` above.
>
> **Q3: Timestep Ablation**
>
> **Answer:** We thank the reviewer for this suggestion and agree that the wording can be made more precise. Our intent was not to present **20 denoising steps** as a standalone methodological novelty, but as an **empirical efficiency–quality operating point** enabled by UniDFlow’s discrete flow-matching formulation.
>
> | Steps | Text Gen | GenEval | DPGBench | ImgEdit | Inf Time |
> |--:|--:|--:|--:|--:|--:|
> | 10    | 81.23    | 0.94    | 90.99    | 4.19 | **4.29**  |
> | 20    | 82.85    | 0.95    | 91.19    | 4.24 | 5.52 |
> | 50    | **83.57**    | **0.95**    | **91.34**    |  **4.33**  | 8.43 |
>
> As shown in the table, increasing the denoising steps from 20 to 50 brings only modest quality gains: Avg(Text Gen) improves from **82.85** to **83.57**, DPGBench from **91.19** to **91.34**, and ImgEdit Bench from **4.24** to **4.33**, while 10 steps clearly underperform (**81.23 / 90.99 / 4.19**). This indicates that **20 steps already capture most of the achievable quality**, making it a practical operating point for strong generation and editing performance.
>
> We will revise the paper to make this point explicit and present the denoising-step study as supporting evidence for the efficiency of the proposed formulation.
>
> **Q4: Figure Confusion**
>
> **Answer:** We will revise the figures to more clearly separate the three stages, indicate frozen vs. trainable modules, and explicitly label the roles of the understanding adapter, generation adapter, and routing/alignment components. We welcome any additional suggestions from the reviewer to further improve our figure design.
>
> **Q5: Similarities with BAGEL**
>
> **Answer:** While both aim at unified multimodal modeling, BAGEL uses a hybrid AR–diffusion design, whereas UniDFlow adopts a single discrete flow-matching interface for both text and image outputs. Our motivation is that hybrid frameworks can suffer from objective mismatch and interference between understanding and generation, which UniDFlow addresses through unified DFM, task-specialized adapters, dynamic routing, and multimodal preference alignment.
>
> Empirically, UniDFlow consistently outperforms BAGEL on understanding, generation on GenEval and DPGBench; and on editing on  IMGEDIT and EMU-EDIT. Our ablations further show where these gains come from. Removing mRef-DPO reduces performance to 0.86 / 86.23 / 4.05, below BAGEL on both GenEval and DPGBench, while removing reflection gives 0.89 / 87.57 / 4.14, with DPGBench also slightly below BAGEL (0.88 / 87.74 / 3.20). This suggests that UniDFlow’s advantage comes from the combination of unified discrete flow matching and task decoupling, with mRef-DPO and reflection supervision being especially important for the final gains in fine-grained generation and editing.
>
> **Q6: Scaling to larger models**
>
> **Answer:** We thank the reviewer for this suggestion. We have already evaluated UniDFlow across four Qwen3-VL backbone sizes (0.6B, 4B, 8B, and 14B), as reported in the model-size ablation in Table 4. Performance improves monotonically across all benchmarks as the backbone scales from 0.6B to 14B, with especially clear gains in text generation and compositional generation, showing that UniDFlow scales effectively without architectural changes.
>
> Notably, even the smallest UniDFlow remains competitive with much larger baselines such as BAGEL (7B+7B), highlighting the efficiency of our design at compact scales. In addition, Figure 10 shows that our largest variant achieves the highest inference throughput among comparable models, indicating that these scaling gains do not come at the cost of runtime efficiency. In the revision, we will present this ablation as a standalone table with clearer discussion.

---

> > ### Author Rebuttal · Reviewer_vGVy · 2026-04-03
> >
> > Thank you for the detailed rebuttal. My main concerns have been addressed.

---

> > > ### Author Response · Authors · 2026-04-03
> > >
> > > Thank you for the follow-up and for indicating that all concerns are fully resolved. We greatly appreciate your continued recommendation for acceptance. We would be very grateful if you might reconsider whether the numerical score could be updated to better align with your final assessment. We, of course, respect your judgment either way, and sincerely appreciate your time and feedback.

---

### Official Review · Reviewer_CwGy · 2026-03-11

**Soundness:** 3
**Presentation:** 3
**Significance:** 3
**Originality:** 3
**Overall Recommendation:** 4
**Confidence:** 3

**Summary:**

This paper proposes UniDFlow, a unified discrete flow matching framework for multimodal understanding, generation, and editing. The method decouples understanding and generation through task-specific LORA adapters, aiming to mitigate objective interference and representation entanglement. In addition, the authors introduce a reference-based multimodal preference alignment strategy, which optimizes relative outcomes under identical conditioning to improve faithfulness and controllability without requiring large-scale retraining. Experiments show that the proposed framework achieves SOTA performance across multiple benchmarks.

**Compliance With Llm Reviewing Policy:**

Affirmed.

**Final Justification:**

The author has addressed my concerns, and I will maintain my positive rating.

**Key Questions For Authors:**

Please see the above weaknesses.

**Limitations:**

Yes.

**Strengths And Weaknesses:**

### Strengths

(1) The paper addresses the problem of designing unified multimodal models for both understanding and generation, which is an important and actively studied research direction.

(2) The motivations behind the proposed components are generally well justified, and the implementations appear relatively simple and intuitive.

(3) The paper is well written and reader-friendly. The inclusion of many illustrative figures helps clarify the proposed methods and visualize the model outputs.

### Weaknesses

Q1: The paper models text prediction as a discrete denoising process, which requires multiple iterative steps. It would be helpful to clarify what advantages this formulation brings. In particular, how does this approach compare with the widely used next-token prediction paradigm in terms of modeling capability, efficiency, or generation quality?


Q2: Several proposed components appear related to existing methods, but their differences and connections are not sufficiently discussed. For example, the proposed MoRA module seems related to Mixture-of-Experts architectures and LoRA-style parameter-efficient adaptation. A clearer explanation of how MoRA differs from these techniques would improve clarity.
Similarly, the use of RMSNorm raises questions about how it differs from Adaptive Layer Normalization used in DiT-style architectures.
Providing a clearer comparison would help readers better understand the novelty of the proposed design.

Q3: The paper repeatedly mentions that the method builds upon a pretrained vision–language transformer, but the specific model used (Qwen3) is only mentioned later in Section 4.4 during the model size ablation study. This information should be introduced earlier in the paper to improve clarity.
In addition, it would be helpful to explain why Qwen3 is used instead of Qwen3-VL, which appears to be a more natural choice for multimodal tasks.

---

> ### Author Rebuttal · Authors · 2026-03-30
>
> We sincerely appreciate the reviewer’s thoughtful and encouraging feedback. We are glad that the reviewer recognized the importance of unified multimodal modeling, the intuitive design of our method, and the clarity of our presentation.
>
> **Q1: Comparison with new paradigm**
>
> **Answer:** We thank the reviewer for this important question. We provide a direct comparison between discrete diffusion and alternative generation paradigms below:
>
> | Methods | Modeling Type | Text Gen | GenEval | DPGBench| ImgEdit Bench  |
> |-|-|-:|--:|--:|-:|
> | Show-O2 | Autoregressive + Flow | 75.52 | 0.88 | 86.63  | - |
> | VARGPT  | Autoregressive  | 74.37 | 0.89 | 85.59 | - |
> | Ours    | Discrete Diffusion | **82.85**| **0.95**   | **91.19** | **4.24**|
>
> UniDFlow's discrete diffusion formulation offers three key advantages over the next-token prediction paradigm.
> First, bidirectional context: unlike autoregressive models that decode left-to-right with causal masking, our discrete flow matching operates with full bidirectional self-attention, enabling the model to condition each token prediction on the complete multimodal context. This is critical for compositional generation tasks (GenEval +6.7% over Show-O2, +6.0% over VARGPT) where global consistency across attributes, objects, and spatial relations must be maintained simultaneously.
>
> Second, unified objective without modality mismatch: hybrid AR+Flow approaches (e.g., Show-O2) couple cross-entropy decoding for text with flow-based regression for images, creating mismatched optimization landscapes that lead to unstable joint training (as discussed in Section 1, Limitation 1). UniDFlow applies a single discrete flow-matching objective (Eq. 1) across both text and visual tokens, eliminating this interference and yielding +7.33 Avg Text Gen improvement over Show-O2.
>
> Third, efficient parallel decoding: our formulation requires only 20 denoising steps (Section 1) with parallel token prediction at each step, whereas autoregressive generation scales linearly with sequence length, a significant bottleneck for high-resolution visual token sequences.
>
> The consistent improvements across all benchmarks, including the ability to perform image editing (ImgEdit 4.24, unavailable for AR baselines), validate that discrete diffusion provides a more effective unified interface for joint multimodal understanding, generation, and editing. We will add this comparison to the revised manuscript.
>
> **Q2: Explanation of how MoRA differs from these techniques**
>
> **Answer:**  We thank the reviewer for raising this point and agree that the differences from related techniques should be stated more explicitly.
>
> MoRA differs from both conventional MoE and standard LoRA. Unlike conventional MoE, which typically introduces multiple full-capacity experts with sparse token dispatch, MoRA does not route tokens to separate experts or increase model capacity in that way. Instead, it keeps the pretrained backbone frozen and uses a lightweight router to compute a hidden-state-dependent mixing coefficient that dynamically interpolates between two task-specific low-rank adapters,\Delta\theta_u for understanding and \Delta\theta_g​ for generation. Compared with a standard shared LoRA, which uses a single low-rank adapter to modify the frozen backbone, MoRA avoids forcing one adapter to absorb both objectives and instead enables dynamic composition of specialized adapters.
>
> This design is motivated by the objective-interference problem in unified multimodal training. Our ablations in Table 4 show that both removing MoRA and collapsing the model into a single shared LoRA lead to worse performance, indicating that dynamic composition of specialized adapters is important rather than mere parameter sharing.
>
> Likewise, our TSG-RMSNorm should be viewed as a backbone-preserving time-conditioning mechanism rather than a direct reuse of DiT-style AdaLN: instead of replacing the normalization design or injecting timestep information into attention/MLP activations, we modulate the pretrained RMSNorm parameters with zero-initialized timestep-dependent scale/bias terms, so the model exactly recovers the pretrained backbone at initialization and adapts to diffusion time with minimal disturbance to the original representation geometry. For an ablation comparing TSG-RMSNorm with AdaLN, please see our response to reviewer `ZmRr`. We will revise the paper to make these distinctions explicit.
>
> **Q3: Confusion between Qwen-3 and Qwen-3-VL**
>
> **Answer:**  We thank the reviewer for pointing this out and apologize for the inconsistency. The backbone used throughout the paper is Qwen3-VL, not the text-only Qwen3 model, and we agree this should be stated clearly much earlier in the manuscript. In the final revision, we will correct the naming throughout, explicitly introduce Qwen3-VL at the beginning of the method section, and We will also release the code and checkpoints to ensure the exact setup is fully reproducible.

---

> > ### Author Rebuttal · Reviewer_CwGy · 2026-04-03
> >
> > The author has addressed my concerns, and I will maintain my positive rating.

---

> > > ### Author Response · Authors · 2026-04-03
> > >
> > > Thank you for the follow-up and for indicating that all concerns are fully resolved. We greatly appreciate your continued recommendation for acceptance. We would be very grateful if you might reconsider whether the numerical score could be updated to better align with your final assessment. We, of course, respect your judgment either way, and sincerely appreciate your time and feedback.

---

### Official Review · Reviewer_ZmRr · 2026-03-11

**Soundness:** 3
**Presentation:** 3
**Significance:** 3
**Originality:** 4
**Overall Recommendation:** 4
**Confidence:** 4

**Summary:**

This paper presents UniDFlow, a unified discrete flow-matching framework that integrates multimodal understanding, high-fidelity image generation, and instruction-guided image editing within a single architecture, addressing the core limitation that existing multimodal systems are largely siloed—vision-language models excel at reasoning but lack native generation capacity, while diffusion models dominate visual synthesis with insufficient semantic grounding, and prior unified frameworks suffer from objective mismatch, task entanglement, and poor editing controllability.
Built on a frozen pre-trained vision-language backbone, UniDFlow uses a parameter-efficient three-stage training pipeline with decoupled task-specific low-rank adapters to avoid interference between understanding and generation tasks. Its key contributions include: 1) Time-Step Guided RMSNorm, which injects diffusion time conditioning without disrupting the pre-trained model’s learned priors; 2) A lightweight dynamic Mixture-of-LoRA Routing module to unify understanding and generation capabilities; 3) mRef-DPO, a novel reference-guided multimodal preference alignment method to boost the faithfulness and controllability of generation and editing.

**Compliance With Llm Reviewing Policy:**

Affirmed.

**Final Justification:**

The authors' responses have resolved my concerns. This is a well-executed study, and I will keep my positive score.

**Key Questions For Authors:**

See weakness.
I hope the authors can provide further in-depth analysis and supplementary experimental validations for them, or clarify whether my understandings exist any inaccuracies.

**Limitations:**

yes

**Strengths And Weaknesses:**

Strengths:
1) This paper designed UniDFlow framework integrates multimodal understanding, text-to-image generation and instruction-guided image editing in a single architecture based on the discrete flow matching objective, solving the core problems of mismatched training objectives and representation entanglement between understanding and generation in existing unified models. Its three-stage decoupled training enables parameter-efficient tuning via independent low-rank adapters, preserves the reasoning ability of pre-trained vision-language models, and avoids the high cost and capability degradation of full-model fine-tuning.


2) The innovative reference-guided multimodal preference alignment method (mRef-DPO) effectively improves the instruction faithfulness and controllability of generation and editing by optimizing relative preferences under the same instruction and visual reference, instead of isolated scalar rewards, providing a new idea for the alignment optimization of multimodal models.


3) The experimental verification is comprehensive and sufficient. UniDFlow achieves SOTA on 8 mainstream benchmarks covering understanding, generation and editing, and its 4B-parameter version significantly outperforms baseline models with several times more parameters. Systematic ablation experiments are conducted to verify the effectiveness of core components, and the impacts of key factors such as model scale and tokenizer are analyzed, with high reference value.

Weaknesses:
1) TSG-RMSNorm serves as a fundamental component for the diffusion-based generation of textual content in this work, yet no corresponding ablation experiments are provided for it. It would be valuable to compare it with other alternative time-injection methods like (1) Directly adding time embeddings (scaled by a learnable weight matrix initialized to zero) to the token embeddings; (2) Freezing the backbone of the VLM and injecting time-step information via an additional dedicated adapter module.
If the overall training cost of such comparative experiments is prohibitively high, the authors are requested to supplement with lightweight experimental results to validate the effectiveness of TSG-RMSNorm.

2) The attention maps presented in Figure 2 are insightful, and further investigations into this phenomenon would be greatly appreciated. For instance, does this precise attention pattern emerge only after Stage 3 training? Alternatively, is it the integration of LoRA_img and Ref-DPO that guides the model to focus more on instruction-critical regions during generation? Additionally, it would be valuable to verify whether this enhanced attention localization is the key factor contributing to the model’s superior performance.

3) For the reflection mechanism designed for reasoning-based editing, only the performance degradation after its removal is verified in the ablation experiment. The impact of the quality of reflection sequences on editing effect is not further analyzed, nor is the difference in the role of the sequences in editing tasks of different complexities explored, resulting in insufficient excavation of the internal logic of the mechanism.

4) Some typo: the score of UniDFlow on DPGBench is 91.19 in Table 2, but 91.91 in the ablation experiment of Table 4 for the same model and benchmark.

---

> ### Author Rebuttal · Authors · 2026-03-30
>
> We thank the reviewer for the highly positive assessment of our work. We especially appreciate the recognition of UniDFlow’s unified design, the novelty of mRef-DPO, and the comprehensiveness of our empirical evaluation.
>
> **Q1: Time Step Guidance**
>
> Answer: We thank the reviewer for this constructive suggestion. We have conducted the requested ablation comparing TSG-RMSNorm (TSGN) against two alternative time-injection strategies: (1) In-context token concatenation, which diminished representation, and (2) AdLN, which replaces pretrained RMSNorm parameters entirely with time-conditioned affine transforms
>
> | Guidance        | Text Gen | GenEval | DPGBench | ImgEdit Bench | Inf. time |
> |-------------------------|---------:|--------:|---------:|--------------:|-----------:|
> | In-Context with tokens  | 82.11    | 0.94    | 90.09    | 4.15          | 5.89 |
> | AdLN                    | 81.53    | 0.93    | 88.75    | 4.05          | 5.78  |
> | TSGN                    | **82.85**    | **0.95**    | **91.19**    | **4.24** | **5.52** |
>
> In-context token concatenation is simple, but it lacks per-layer granularity and its time signal weakens as sequence length grows (up to 4096 tokens in Stage III), resulting in lower DPGBench (−1.10) and ImgEdit (−0.09) scores. AdLN performs worse because it overwrites the learned normalization statistics of the frozen Qwen3-VL backbone and disrupts the reasoning priors we aim to preserve.
>
> By contrast, TSGN modulates existing RMSNorm parameters with zero-initialized multiplicative $(1+s_\ell(t))$ and additive $b_\ell(t)$ terms, exactly recovering the pretrained model at initialization. This preserves pretrained hidden-state structure while providing per-layer, time-dependent conditioning, yielding the best balance between preserving backbone reasoning priors, with only negligible overhead beyond the LoRA adapters. We will include this ablation.
>
> **Q2: Attention Pattern, integration of LoRA_img and Ref-DPO**
>
> **Answer:** The attention maps in Fig. 2 are from the fully trained model after Stage III. Our stage-wise analysis shows that coherent, semantically meaningful attention already emerges after Stage I, indicating that UniDFlow inherits localization ability from the pretrained Qwen3-VL backbone. Stage II, via LoRA_img, adds instruction-conditioned visual token generation, while Stage III further sharpens this behavior through mRef-DPO, improving cross-modal grounding and focus on instruction-critical regions. Thus, the observed localization is not unique to Stage III, but progressively strengthened across stages. We believe this improved localization contributes to stronger fine-grained instruction following and editing performance. In the camera-ready version, we will make this clearer by adding attention map comparisons for Qwen3-VL, UniDFlow-Stage-I, and UniDFlow-Stage-III
>
> **Q3: sequences in editing tasks of different complexities**
>
> **Answer:** We thank the reviewer for this insightful suggestion. Since reflection is generated in parallel with the edit output, we study its role through a complexity-controlled analysis over add/remove, reasoning-based, scene-text, and multi-object editing. The results show that reflection consistently improves PSNR/SSIM, reduces structure distance, and increases CLIP similarity for both edited and preserved regions, with especially clear gains on reasoning-based and multi-object edits.
>
> | Setting       | Complexity              | PSNR  | SSIM  | Structure-Distance | Clip Score(Edited Part) | Clip Score (Unedited part) |
> |---------------|-------------------------|------:|------:|--------------------------:|------------------------:|---------------------------:|
> | Reflection    | Add Remove              | 31.43 | 0.856 | 9.12                      | 26.67                   | 25.05                      |
> | ,,    | Resoning Based Editing  | 27.82 | 0.869 | 7.45                      | 27.53                   | 24.11                      |
> | ,,    | Scene Text Editing      | 30.11 | 0.891 | 10.43                     | 31.44                   | 27.62                      |
> | ,,    | Multi Object Editing    | 31.45 | 0.847 | 9.52                      | 28.82                   | 25.01                      |
> | WO Reflection | Add Remove   | 29.44 | 0.833 | 11.33                     | 24.44                   | 23.37                      |
> | ,, | Resoning Based Editing  | 23.67 | 0.849 | 10.02 | 23.12 | 23.78  |
> | ,, | Scene Text Editing      | 27.52 | 0.871 | 12.02  | 27.72 | 25.28  |
> | ,, | Multi Object Editing    | 28.89 | 0.825 | 11.78 | 25.17 | 21.75 |
>
> This indicates that reflection primarily helps the model localize the correct edit region. We will include this quantitative analysis in the final revision
>
> **Q4: Typo**
>
> **Answer:** We thank the reviewer for carefully spotting this inconsistency. This discrepancy is due to a typo in the manuscript, and the reported DPGBench score is correct (91.19). We will correct the number in Table.

---

> > ### Author Rebuttal · Reviewer_ZmRr · 2026-04-05
> >
> > The authors' responses have resolved my concerns. This is a well-executed study, and I will keep my positive score.

---

> > > ### Author Response · Authors · 2026-04-05
> > >
> > > Thank you for the follow-up and for indicating that all concerns are fully resolved. We greatly appreciate your continued recommendation for acceptance. We would be very grateful if you might reconsider whether the numerical score could be updated to better align with your final assessment. We, of course, respect your judgment either way, and sincerely appreciate your time and feedback.

---

### Official Review · Reviewer_DHXZ · 2026-03-13

**Soundness:** 3
**Presentation:** 3
**Significance:** 2
**Originality:** 2
**Overall Recommendation:** 4
**Confidence:** 3

**Summary:**

This paper studies the tension between multimodal understanding and generation in a unified model. The authors propose a shared framework with separate adapters for the two capabilities and a router to coordinate them. The training recipe is staged, and notably includes a DPO-based alignment step, which is used to further improve generation behavior without fully sacrificing understanding performance. The main contribution is thus a unified but functionally decoupled multimodal architecture for balancing these two objectives.

**Compliance With Llm Reviewing Policy:**

Affirmed.

**Final Justification:**

The authors' rebuttal has successfully addressed my concerns. Therefore, I will maintain my initial positive score.

**Key Questions For Authors:**

Same as weaknesses

**Limitations:**

The discussion could be more explicit about the overall system complexity, training/data cost, and the resulting reproducibility barrier. It would also help to briefly comment on potential risks inherited from large-scale multimodal generation systems, such as biased or hallucinated outputs.

**Strengths And Weaknesses:**

Strengths

1. The paper addresses an important  problem in multimodal modeling——the tension between understanding and generation within a unified model.
2. The proposed design—separate adapters for the two capabilities with a router on top—is intuitive and practically motivated, and the inclusion of a staged training recipe with a DPO-based alignment step is sensible.
3. The paper is clearly written and the main idea is easy to follow

Weaknesses
1. The technical novelty is somewhat moderate, as the method mainly combines existing ingredients in a structured way rather than introducing a fundamentally new modeling paradigm.
2. The overall framework is also fairly complex, with multiple components and training stages, which raises the bar for reproduction and makes it harder to disentangle where the gains truly come from.

---

> ### Author Rebuttal · Authors · 2026-03-30
>
> We sincerely thank the reviewer for the positive assessment and recognizing the importance of the problem. We appreciate the reviewer's recognition of the value and motivation for our unified design.
>
> **Q1: The technical novelty**
>
> **Answer:**  Thank you for this comment. We agree that UniDFlow is not positioned as a completely new generative paradigm. Rather, the technical novelty lies in a new unified formulation and training recipe for multimodal understanding, generation, and editing under one discrete low-matching VLM.
>
> The contributions of our work are:
>
> (1) We introduce a single DFM-based interface with a three-stage optimization recipe that first aligns diffusion-style understanding with a KL anchor to the pretrained autoregressive VLM, then learns generation separately, and only afterward performs joint multimodal alignment. This is designed to avoid the objective interference that occurs under naive parameter sharing.
>
> (2) We propose TSG-RMSNorm, a time-conditioning mechanism that injects diffusion time by modulating RMSNorm scales rather than perturbing hidden activations, thereby preserving pretrained representations during diffusion adaptation;
>
> (3) We introduce mRef-DPO, a reference-anchored multimodal preference objective that jointly aligns text outputs and visual edits against a frozen reference policy, enabling grounded and faithful editing rather than text-only preference tuning.
>
> (4) We introduce MoRA, a lightweight hidden-state-dependent routing strategy that dynamically composes understanding and generation adapters instead of relying on a static or shared LoRA.
>
> UniDFlow unifies multimodal reasoning, thinking-based text-to-image generation, and instruction-driven image editing within a single framework, achieves state-of-the-art performance across eight benchmarks, and exhibits strong zero-shot generalization, in-context image generation, reference based editing, and compositional generation.
>
> **Q2: Framework Complexity**
>
> **Answer:** Thank you for this important comment. We agree that UniDFlow includes multiple components, and we will clarify the design and training recipe more clearly in the revision. To improve reproducibility, we plan to release the full training code, stage-wise checkpoints, configuration files, and data-processing details, so both the final system and intermediate variants can be reproduced.
> At the same time, we would like to emphasize that the framework is intentionally parameter-efficient. Task adaptation uses lightweight LoRA modules, and the final joint stage adds only a small router rather than updating the full model. Because jointly training understanding, generation, and editing objectives in a single model can cause strong objective interference, the stage-wise design is what enables stable convergence. We therefore view UniDFlow as a modular and inspectable training recipe rather than an opaque pipeline.
>
> We also agree that it is important to disentangle where the gains come from. This is exactly why Table 4 includes ablations over model size, tokenizer, architectural choices, loss terms, and the Stage III alignment strategy. The results show that removing the understanding adapter, generation adapter, router, or multimodal preference terms consistently hurts performance. For example, removing MoRA reduces performance to 80.67 / 0.93 / 89.88 / 4.11, while replacing it with a single shared LoRA further lowers performance to 79.92 / 0.90 / 89.12 / 4.08, compared with 82.85 / 0.95 / 91.91 / 4.24 for the full model. Similarly, replacing mRef-DPO with vanilla DPO or uni-GRPO also lowers performance. In other words, the gains do not come from an opaque stack of add-ons, but from components whose roles are individually validated in the ablations.
>
> **Q3 Limitations:**
>
> **Answer:** We appreciate this point and will make the discussion more explicit. UniDFlow employs a modular three-stage training recipe because jointly optimizing understanding, generation, and editing in one pass causes objective interference, but we agree this increases system complexity and raises the reproduction bar. To address this, we will release the complete training code, configs, data-processing details, and checkpoints for each stage, and report the stage-wise data/training budget transparently, approximately 0.6T, 1.2T, and 1.8T image-text tokens for Stages I–III, respectively (totaling 3.6T) so others can reproduce both the final model and intermediate variants in a controlled way.
>
> We will also state more clearly that, although UniDFlow is parameter-efficient at adaptation time and improves faithfulness and controllability through reference-based multimodal preference alignment, it still inherits the cost and limitations of large-scale multimodal generation systems, including the risk of biased, hallucinated, or misleading outputs and misuse for deceptive image manipulation, which motivates safeguards such as moderation, bias evaluation, and provenance or watermarking.

---

> > ### Author Rebuttal · Reviewer_DHXZ · 2026-04-02
> >
> > The author has addressed my concerns, and I will maintain my recommendation for acceptance. I hope the author will release the code.

---

> > > ### Author Response · Authors · 2026-04-02
> > >
> > > Thank you for the follow-up and for indicating that all concerns are fully resolved. We greatly appreciate your continued recommendation for acceptance. We would be very grateful if you might reconsider whether the numerical score could be updated to better align with your final assessment. We, of course, respect your judgment either way, and sincerely appreciate your time and feedback.

---

### Decision · Program_Chairs · 2026-04-30

**Decision:**

Accept (regular)

**Comment:**

The paper receives consistently positive evaluations from all reviewers, who agree that it addresses an important problem in unified multimodal understanding, generation, and editing. The proposed framework, based on task-specific adapters, a routing mechanism, and staged training, is considered technically sound, well motivated, and clearly presented. Reviewers particularly highlight the strong and comprehensive empirical results across multiple benchmarks. The main concerns focus on the degree of novelty (incremental), the complexity of the multi-stage framework, and the need for clearer analysis or positioning of certain components. However, the authors’ rebuttal effectively addresses these points by providing additional experiments and clarifications. All reviewers explicitly acknowledge that their concerns have been resolved and maintain positive recommendations.